# Independent regulation of mitochondrial DNA quantity and quality in *Caenorhabditis elegans* primordial germ cells

**Aaron ZA Schwartz[1,2], Nikita Tsyba[3], Yusuff Abdu[1,2], Maulik R Patel[3,4,5], Jeremy Nance[1,2]***

[1]Department of Cell Biology, NYU Grossman School of Medicine, New York, United States; [2]Skirball Institute of Biomolecular Medicine, NYU Grossman School of Medicine, New York, United States; [3]Department of Biological Sciences, Vanderbilt University, Nashville, United States; [4]Department of Cell and Developmental Biology, Vanderbilt University School of Medicine, Nashville, United States; [5]Diabetes Research and Training Center, Vanderbilt University School of Medicine, Nashville, United States

**Abstract** Mitochondria harbor an independent genome, called mitochondrial DNA (mtDNA), which contains essential metabolic genes. Although mtDNA mutations occur at high frequency, they are inherited infrequently, indicating that germline mechanisms limit their accumulation. To determine how germline mtDNA is regulated, we examined the control of mtDNA quantity and quality in *C. elegans* primordial germ cells (PGCs). We show that PGCs combine strategies to generate a low point in mtDNA number by segregating mitochondria into lobe-like protrusions that are cannibalized by adjacent cells, and by concurrently eliminating mitochondria through autophagy, reducing overall mtDNA content twofold. As PGCs exit quiescence and divide, mtDNAs replicate to maintain a set point of ~200 mtDNAs per germline stem cell. Whereas cannibalism and autophagy eliminate mtDNAs stochastically, we show that the kinase PTEN-induced kinase 1 (PINK1), operating independently of Parkin and autophagy, preferentially reduces the fraction of mutant mtDNAs. Thus, PGCs employ parallel mechanisms to control both the quantity and quality of the founding population of germline mtDNAs.

*For correspondence:
jeremy.nance@med.nyu.edu

**Competing interest:** The authors declare that no competing interests exist.

## Editor's evaluation

Mitochondria have their own DNA, which is much more likely to gain mutations (due to error-prone DNA polymerase). It is widely appreciated that there are quality control mechanisms such that functional mitochondria are passed from one generation to the next. This manuscript presents important progress in the field, describing how the *C. elegans* germline may remove mitochondria by creating bottlenecks as well as selectively removing non-functional mitochondria. Building upon the authors' previous finding that the *C. elegans* primordial germ cells (PGCs) shed much of their cytoplasm during embryogenesis through 'cannibalism', they now describe that a bulk of mitochondria are removed from PGCs through this process. Although some of the phenotypes described in the manuscript are relatively mild, the evidence is compelling, supporting their conclusions.

**eLife digest** Mitochondria are the powerhouses of every cell in our bodies. These tiny structures convert energy from the food we eat into a form that cells are able to use. As well as being a separate organ-like structure within our cells, mitochondria even have their own DNA. Mitochondrial DNA contains genes for a small number of special enzymes that allow it to extract energy from food. In contrast, the rest of our cells' DNA is stored in another structure called the nucleus.

Mitochondrial and nuclear DNA are also inherited differently. We inherit nuclear DNA from both our mother and father, but mitochondrial DNA is only passed down from our mothers. During reproduction, maternal DNA (including mitochondrial DNA) comes from the egg cell, which combines with sperm to produce offspring.

Defects, or mutations, in mitochondrial genes often lead to mitochondrial diseases. These have a severe impact on health, especially during the very first stages of life. The lineage of precursor cells that gives rise to egg cells is thought to protect itself from mitochondrial mutations, but how it does this is still unclear. Schwartz et al. therefore set out to determine what molecular mechanisms preserve the integrity of mitochondrial DNA from one generation to the next.

To address this question, *C. elegans* roundworms were used, as they are easy to manipulate genetically, and since they are small and transparent, their cells – as well as their mitochondria – are also easily viewed under a microscope. Tracking mitochondria in the worms' egg precursor cells (also called primordial germ cells, or PGCs) revealed that PGCs actively removed excess mitochondria. The PGCs did this either by internally breaking down mitochondria themselves, or by moving them into protruding lobe-like structures which surrounding cells then engulfed and 'digested'.

Further genetic studies revealed that the PGCs also directly regulated the quality of mitochondrial DNA via a mechanism dependent on the protein PINK1. In worms lacking PINK1, mutant mitochondrial DNA remained in the PGCs at high levels, whereas normal worms successfully reduced the mutant DNA. Thus, the PGCs used parallel mechanisms to control both the quantity and quality of mitochondria passed to the next generation.

These results contribute to our understanding of how organisms safeguard their offspring from inheriting mutant mitochondrial DNA. In the future, Schwartz et al. hope that this knowledge will help us treat inherited mitochondrial diseases in humans.

## Introduction

Mitochondria contain multiple copies of a small genome called mitochondrial DNA (mtDNA), which includes several genes essential for oxidative phosphorylation (*Fu et al., 2020*). Compared to nuclear DNA, mtDNA has a high mutation rate and is repaired inefficiently (*Fu et al., 2020*). The mtDNAs with deleterious mutations are found together with complementing wild-type mtDNAs in a state called heteroplasmy. Deleterious mtDNA mutations can lead to mitochondrial disease if present at sufficiently high heteroplasmy – a condition that is estimated to affect ~1 in 5000 individuals and has no known cure (*Gorman et al., 2015*).

Mitochondrial DNA replicates independently from nuclear DNA and has a distinct mode of inheritance. During cell division in most cell types, each daughter inherits a stochastic subset of mitochondria and their mtDNAs. However, embryos inherit their mtDNAs exclusively from the pool present within the oocyte (*Palozzi et al., 2018*). The strict maternal inheritance and high mutation rate of mtDNA raise a potential problem: mtDNA mutations could accumulate over generations, leading to mutational meltdown (*Muller, 1964*). However, relatively few deleterious mutations are transmitted over generations (*Nachman, 1998*), indicating that mtDNA mutations are selected against within the germ line.

Two mechanisms have been proposed to regulate germline mtDNA inheritance. In one mechanism – the mitochondrial bottleneck – mtDNAs are reduced in number within the germline lineage to create a small founding population, which is passed on to the next generation. In theory, genetic bottlenecks allow for the stochastic enrichment or depletion of variant mtDNAs in germ cells, potentially enabling selection against detrimental mtDNA mutations in subsequent generations (*Palozzi et al., 2018*; *Hauswirth and Laipis, 1982*; *Olivo et al., 1983*). In vertebrates, a bottleneck occurs in embryonic primordial germ cells (PGCs) due to the dilution of maternally provided mtDNAs by

reductive embryonic cell divisions, or via the replication of a subset of mtDNA genomes in PGCs (*Cao et al., 2007*; *Cree et al., 2008*; *Floros et al., 2018*; *Jenuth et al., 1996*; *Otten et al., 2016*; *Wai et al., 2008*). It is not known whether germline mtDNA bottlenecks could form through other means.

Alternatively, mitochondria containing high levels of mutant mtDNAs can be eliminated directly from germ cells – a process called purifying selection (*Palozzi et al., 2018*). The mechanistic basis for germline purifying selection has been studied most intensively in the *Drosophila* ovary, where mtDNA mutations are eliminated both by autophagy and selective mtDNA replication (*Chen et al., 2020*; *Lieber et al., 2019*; *Zhang et al., 2019*; *Hill et al., 2014*; *Ma et al., 2014*). Although there is genetic evidence for purifying selection in many species, including humans (*Floros et al., 2018*), it is unknown whether it occurs through the mechanisms identified in flies or if alternative mechanisms for purging mutant germline mtDNAs exist.

## Results

### PGCs eliminate mitochondria through intercellular cannibalism

To identify additional mechanisms of germline mtDNA control, we investigated how mtDNA quantity and quality are regulated in *Caenorhabditis elegans* PGCs. The entire *C. elegans* germ line descends from two PGCs, which are born early in embryogenesis and remain quiescent until early larval stages (*Fukuyama et al., 2006*). Although embryonic PGCs do not divide, they undergo a non-mitotic cellular remodeling process, discarding much of their cell mass and content. Remodeling occurs when PGCs form organelle-filled lobe-like protrusions, which adjacent endodermal cells cannibalize and digest (*Figure 1A*; *Abdu et al., 2016*; *Sulston et al., 1983*). Previously, we showed that PGCs lose much of their mitochondrial mass in the process of lobe cannibalism, suggesting that one role of this remodeling event could be to eliminate PGC mitochondria in bulk (*Abdu et al., 2016*). As such, lobe cannibalism might provide a novel mechanism for PGCs to adjust their mtDNA quantity and/or quality at the initial stages of germline development.

To begin to test this hypothesis, we used PGC-specific markers of the plasma membrane (PH$_{PLC1\partial1}$::mCherry, 'Mem-mCh$^{PGC}$') and mitochondrial outer membrane (TOMM-20$^{1-54}$::Dendra2, 'Mito-Dendra$^{PGC}$') to follow the distribution of PGC mitochondria during lobe formation and cannibalization in living embryos. Most PGC mitochondria moved into lobes shortly after they formed (*Figure 1B–C*) but a subset returned to the cell body prior to lobe digestion (*Figure 1D and F*, *Figure 1—figure supplement 1A-B*). Cell body mitochondria that are retained in L1 PGCs (*Figure 1E*) represent the founding population present at the onset of larval germline expansion.

The PGC lobe fragments present within endodermal cells colocalize with the lysosomal marker LAMP-1, suggesting that mitochondria within lobes are targeted for destruction and digested (*Abdu et al., 2016*). To test this hypothesis more directly, we visualized the mitochondrial outer membrane marker Mito-Dendra$^{PGC}$, which is pH-sensitive (*Chudakov et al., 2007*) and should be quenched when mitochondria are present within lysosomes. In L1 larvae, Mito-Dendra$^{PGC}$ fluorescence was greatly reduced in cannibalized lobe mitochondria compared to pH-insensitive Mito-mCh$^{PGC}$ (*Shaner et al., 2004*; arrowheads, *Figure 1G–H*), whereas both markers labeled PGC cell body mitochondria robustly (dashed outline, *Figure 1G–H*). We conclude that PGC lobe mitochondria are digested by endodermal cells shortly after lobes are cannibalized, permanently removing them from the mitochondrial pool passed on to L1 larval PGCs.

### Lobe cannibalism and autophagy halve the number of PGC mtDNAs

To determine how elimination of mitochondria by lobe cannibalism affects the pool of germline mtDNAs, we first examined PGC mtDNAs visually. Mitochondrial transcription factor-A (TFAM), a component of the mtDNA nucleoid, is a well-characterized marker of mtDNA (*Garrido et al., 2003*; *Lewis et al., 2016*; *Rajala et al., 2014*). In human cells, individual TFAM nucleoids appear as puncta within the mitochondrial matrix and contain single, or at most a few, mtDNA genomes (*Brown et al., 2011*; *Kukat et al., 2011*). We tagged the *C. elegans TFAM* homolog (*hmg-5*) endogenously with *green fluorescent protein (GFP)*. The TFAM-GFP protein was expressed ubiquitously and formed puncta that localized to mitochondria, consistent with its known binding to mtDNA in *C. elegans* (*Figure 1—figure supplement 2A-B*; *Yang et al., 2022*). Within PGCs, TFAM-GFP puncta were present in both cell body and lobe mitochondria, including those that had been recently cannibalized (arrowhead,

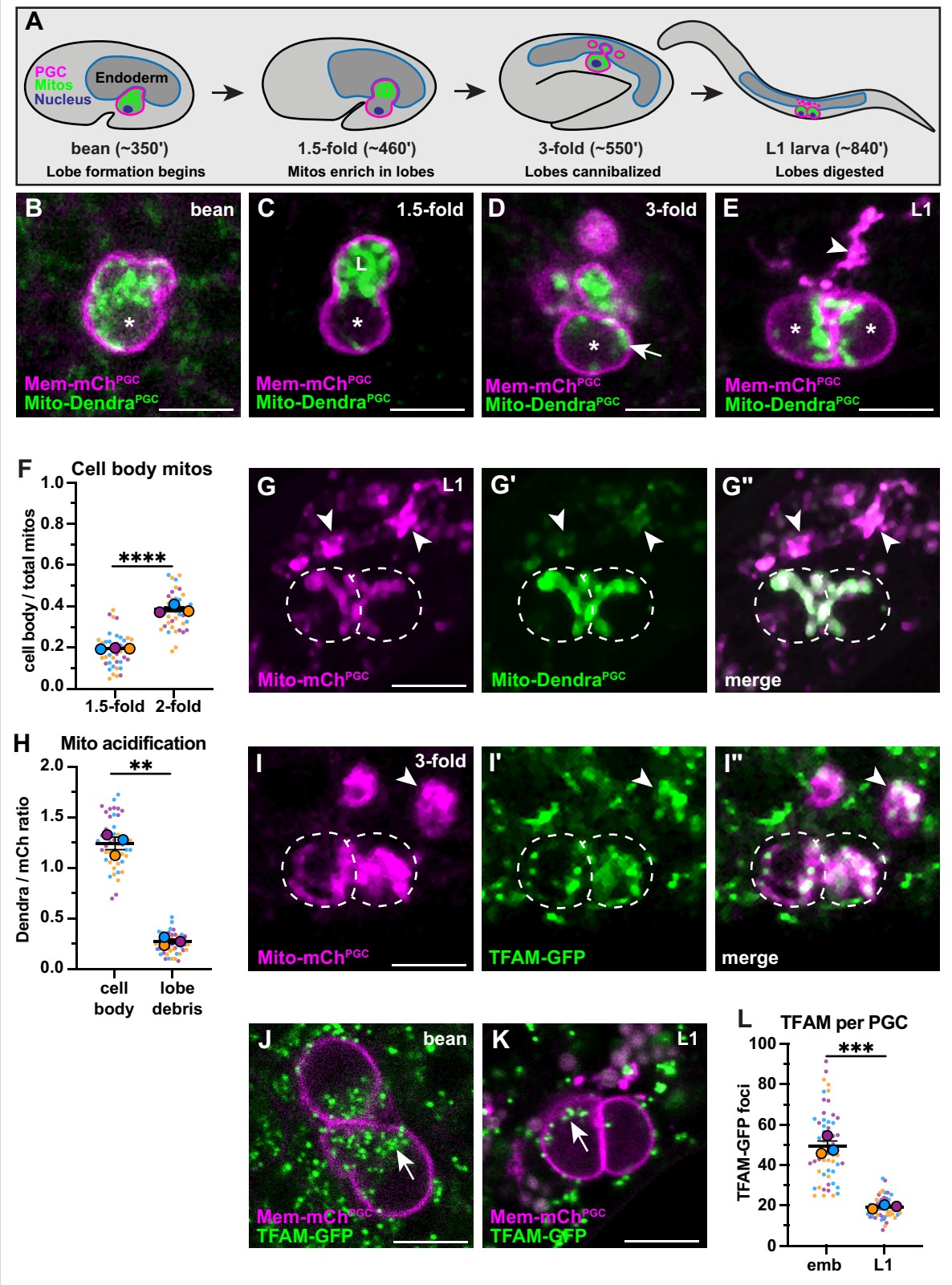

**Figure 1.** Primordial germ cell (PGC) lobe mitochondria and mitochondrial DNAs (mtDNAs) are cannibalized and digested. (**A**) Schematic of PGC lobe formation and cannibalism. Bean stage to threefold embryos, one PGC visible; L1 larva, both PGCs visible. PGCs (magenta), PGC mitochondria (green), and endoderm (blue) are shown. Developmental timepoints are shown as approximate time in minutes post-fertilization at 20–22°C. (**B–E**) Plasma membranes and mitochondria in embryonic PGCs just as lobes form (**B**), in PGCs with lobes (**C–D**), and in L1 larval PGCs after lobes are

*Figure 1 continued on next page*

*Figure 1 continued*

digested (**E**; arrowhead, lobe debris in endoderm). *, nucleus; 'L', lobe. (**F**) Quantification of the mitochondrial fraction within the cell body in 1.5-fold and 2-fold stage PGCs. (**G-G"**) Acidified mitochondria (arrowheads) in digested PGC lobes of L1 larvae. Dashed lines, the outline of PGC cell bodies. (**H**) Quantification of Mito-Dendra$^{PGC}$ over Mito-mCh$^{PGC}$ ratio in L1 PGCs revealing acidification in lobe debris relative to the cell body. (**I-I"**) Mitochondrial transcription factor-A (TFAM)-green fluorescent protein (GFP) puncta within PGC mitochondria, present in both the cell body (dashed outlines) and in recently cannibalized lobes (arrowheads). Due to the movement of threefold embryos within the eggshell, TFAM-GFP appears diffuse. (**J–L**) TFAM-GFP in embryonic (**J**) and L1 larval (**K**) PGCs. (**L**) Quantification of TFAM-GFP foci in embryonic and L1 PGCs. Data in graphs are shown as a Superplot, with individual data points from three independent color-coded biological replicates shown as small dots, the mean from each experiment shown as a larger circle, the mean of means as a horizontal line, and the SEM as error bars. **p≤0.01, ***p≤0.001, ****p≤0.0001, unpaired two-tailed Student's *t*-test (**F,L**) and paired-ratio Student's *t*-test (**H**). Scale bars, 5 µm.

The online version of this article includes the following source data and figure supplement(s) for figure 1:

**Source data 1.** Related to *Figure 1F, H and L*.

**Figure supplement 1.** A subset of primordial germ cell (PGC) mitochondria is retained in the cell body prior to lobe digestion.

**Figure supplement 2.** Mitochondrial transcription factor-A (TFAM)-GFP mitochondrial localization and effect on mitochondrial DNA (mtDNA) copy number.

**Figure supplement 2—source data 1.** Related to *Figure 1—figure supplement 2B-C*.

*Figure 1I*). The number of TFAM-GFP foci decreased more than twofold between embryogenesis and the L1 larval stage (*Figure 1J–L*), suggesting that lobe cannibalism results in a substantial loss of PGC mtDNAs.

To quantify the number of mtDNAs within PGCs, we developed a fluorescence activated cell sorting (FACS) protocol to purify GFP-labeled PGCs from dissociated embryos (before lobe cannibalism), late embryos (after lobe cannibalism), and L1 larvae, which we paired with droplet digital PCR (ddPCR) to count mtDNA molecules per cell (*Figure 2A*, *Figure 2—figure supplement 1*, and *Figure 2—figure supplement 2*). We were able to isolate nearly pure populations of PGCs as determined by live imaging (*Figure 2—figure supplement 2A*) and post-sort analysis (see Methods). Additionally, PGCs isolated from late embryos and L1 larvae were less than half the volume of embryonic PGCs (*Figure 2—figure supplement 2A, B*), indicating that lobe cannibalism had not yet initiated in most of the sorted embryonic PGCs and was complete in late embryonic and L1 PGCs, as expected (*Abdu et al., 2016*).

We determined that each embryonic PGC contained 401 ± 11 mtDNAs (*Figure 2B*), which is 1.2% of the number of mtDNAs we detected in whole early embryos (33,840 ± 1784) (*Figure 1—figure supplement 2C*). The volume of each embryonic PGC in vivo (275 ± 7.2 µm³) is 1.2% of the volume of whole embryos (23,949 ± 175 µm³) (*Figure 2—figure supplement 3*), suggesting that embryonic PGCs inherit their mtDNAs from the pool present at fertilization through reductive embryonic cell divisions. By contrast, late embryonic PGCs, which lacked lobes, contained only 272 ± 8.1 mtDNAs (*Figure 2B*). Sorted L1 larval PGCs contained even fewer mtDNAs (220 ± 12; *Figure 2B*). Together, these data suggest that PGC lobe cannibalism could eliminate a third of the mtDNA molecules that each PGC inherits at its birth, and that an additional mechanism further reduces mtDNA numbers between late embryogenesis and the L1 larval stage. Thus, each PGC in the larval germ line contains roughly half the number that they inherit during embryogenesis.

To directly test whether lobe cannibalism contributes to the loss of mtDNAs in PGCs, we examined PGC mtDNA number in *nop-1* mutants, in which most PGCs fail to form lobes (*Figure 2C–D*; *Maniscalco et al., 2020*). The *nop-1* mutant L1 PGCs retained a significantly higher proportion of embryonic PGC mtDNAs compared to wild type (*Figure 2E and G*). This finding implicates lobe cannibalism in the reduction in mtDNA that occurs as PGCs transition from embryogenesis to the L1 stage.

Autophagy is an additional mechanism by which cells can remove cellular components and organelles, including mitochondria. During autophagy, an autophagosome membrane encapsulates organelles and cytoplasm, subsequently fusing with a lysosome to degrade its contents (*Dikic and Elazar, 2018*). To test if autophagy could be the source of lobe-independent mtDNA reduction in PGCs, we used the pH-discriminating Mito-mCh$^{PGC}$ and Mito-Dendra$^{PGC}$ reporters to observe whether any PGC mitochondria become acidified before lobe cannibalism occurs. We observed one or more large, distinct foci of acidified mitochondria [mCherry(+) Dendra(-)] within many PGCs prior to lobe cannibalism (*Figure 2—figure supplement 4*), and foci were absent in autophagy-defective *atg-18/WD repeat domain phosphoinositide interacting 2 (WIPI2)* mutants (see Figure 5F–G; *Palmisano and*

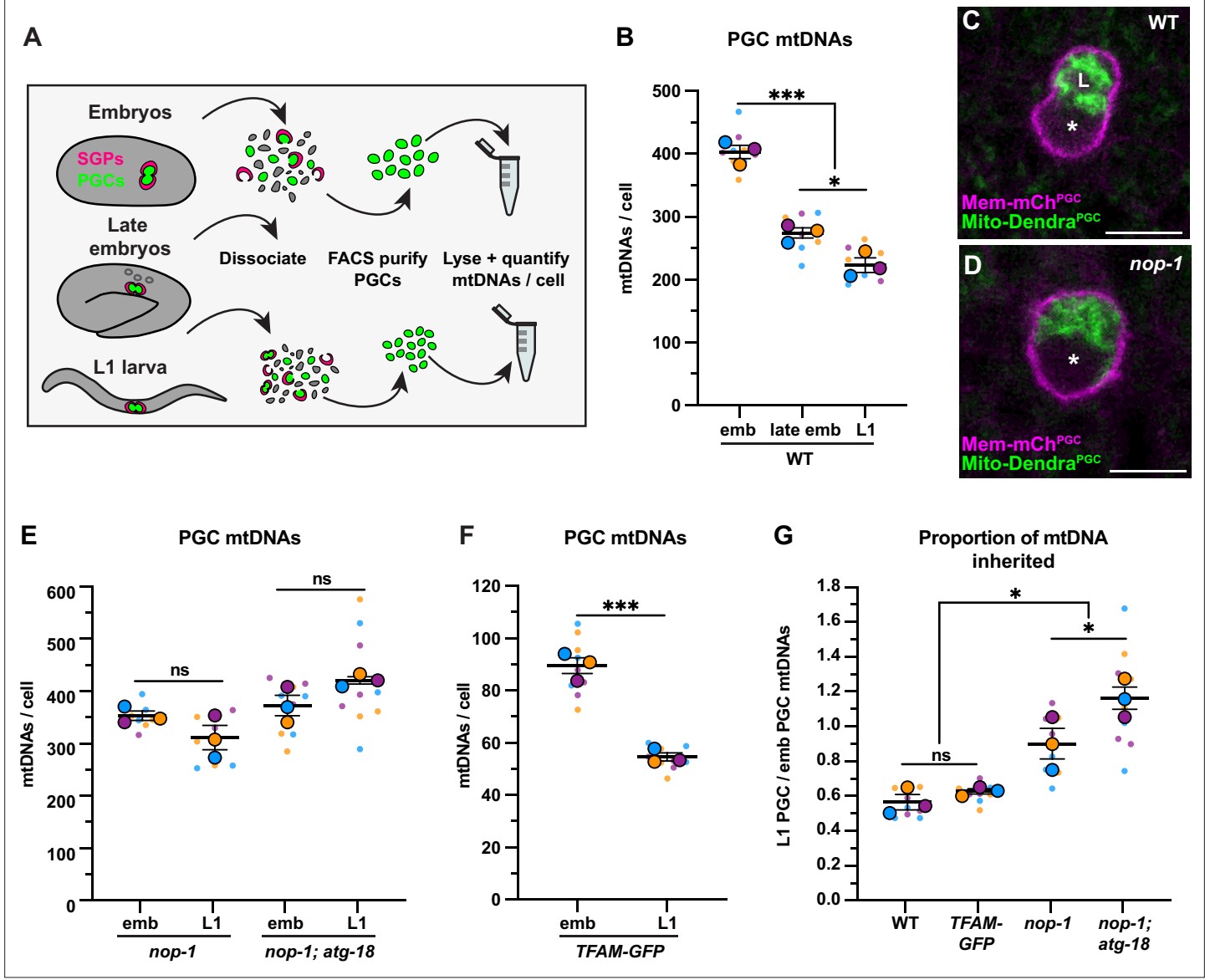

**Figure 2.** Primordial germ cell (PGC) lobe cannibalism and autophagy eliminate a fixed fraction of mitochondrial DNAs (mtDNAs). (**A**) Schematic of fluorescence activated cell sorting (FACS) strategy to isolate PGCs from dissociated embryos and L1 larvae and quantify mtDNAs (see also *Figure 2—figure supplements 1 and 2*). (**B**) Quantification of mtDNA copy number per PGC in wild-type embryos, late embryos, and L1 larvae. (**C–D**) Mitochondria and plasma membrane in wild-type and *nop-1* mutant PGCs. (**E–F**) Quantification of mtDNA copy number per PGC in *nop-1* mutant, *nop-1; atg-18* mutant, and *mitochondrial transcription factor-A (TFAM)-GFP* embryos and L1 larvae. (**G**) The proportion of embryonic PGC mtDNAs inherited by L1 PGCs in wild-type, *TFAM-GFP*, *nop-1* mutants, and *nop-1; atg-18* mutants (from data in B,E, and F). Data in graphs: small dots are three technical replicates of droplet digital PCR (ddPCR) quantification from each of three color-coded biological replicates; the technical replicate mean from each experiment is shown as a larger circle, the mean of means as a horizontal line, and the SEM as error bars. n.s., not significant (p>0.05), *p≤0.05, ***p≤0.001, unpaired one-tailed (**E,G**), and two-tailed (**B,E,F,G**) Student's *t*-tests. Scale bars, 5 μm.

The online version of this article includes the following source data and figure supplement(s) for figure 2:

**Source data 1.** Related to *Figure 2B and E–G*.

**Figure supplement 1.** Embryo and L1 primordial germ cell (PGC) fluorescence activated cell sorting (FACS) purification gating strategy.

**Figure supplement 2.** Fluorescence activated cell sorting (FACS) quality control and droplet digital PCR (ddPCR).

**Figure supplement 2—source data 1.** Related to *Figure 2—figure supplement 2B*.

**Figure supplement 3.** Quantification of whole embryo and primordial germ cell (PGC) volume.

**Figure supplement 3—source data 1.** Related to *Figure 2—figure supplement 3A-B*.

*Figure 2 continued on next page*

*Figure 2 continued*

**Figure supplement 4.** Acidification of a subset of primordial germ cell (PGC) mitochondria.

**Figure supplement 4—source data 1.** Related to *Figure 2—figure supplement 4B*.

*Meléndez, 2019*), suggesting that autophagy may be responsible for lobe-independent degradation of mtDNA in PGCs.

To directly test whether autophagy and lobe cannibalism fully account for mtDNA reduction in PGCs, we sorted embryonic and L1 PGCs from *nop-1; atg-18* double mutants, which are deficient in both lobe cannibalism and autophagy. Notably, mtDNA reduction was entirely prevented in *nop-1; atg-18* L1 PGCs, which inherited a larger proportion of embryonic PGC mtDNAs than *nop-1* single mutants (*Figure 2E and G*). This is consistent with our finding that late embryonic PGCs contain slightly more mtDNAs than L1 PGCs (*Figure 2B*). Together, these results (and data below; see Figure 5A–B) suggest that lobe cannibalism and autophagy are both required for complete mtDNA copy number reduction in PGCs, and indicate that if any mtDNA replication occurs in late embryonic or L1 PGCs, it is insufficient to outpace autophagy-mediated mitochondrial destruction at this stage.

Lobe cannibalism and autophagy could reduce the number of mtDNAs to a fixed number, or alternatively, eliminate a fixed proportion of the mtDNAs present within PGCs regardless of how many are present. To distinguish between these possibilities, we took advantage of the fact that changing TFAM activity can alter mtDNA copy number (*Larsson et al., 1998*; *Sumitani et al., 2011*). Indeed, we found that whole embryos from the *TFAM-GFP* knock-in strain contained significantly fewer mtDNAs (8450 ± 768) than wild type (*Figure 1—figure supplement 2C*), indicating that the GFP tag partially interferes with TFAM function. Using the *TFAM-GFP* strain, we asked how many mtDNAs PGCs eliminate if they are born with a reduced number. The *TFAM-GFP* embryonic PGCs contained 89 ± 3 mtDNAs and *TFAM-GFP* L1 PGCs contained 54 ± 2 mtDNAs (*Figure 2F*). Thus, despite the presence of markedly fewer mtDNAs in the *TFAM-GFP* strain, L1 PGCs still inherit a comparable percentage of the mtDNAs contained within embryonic PGCs (wild-type: 55% and *TFAM-GFP*: 60%; *Figure 2F–G*). Conversely, when embryonic PGCs contained ~25% excess mtDNAs (in the *mptDf2* mtDNA mutant strain), we still observed a twofold reduction in mtDNAs by the L1 stage (see *Figure 4—figure supplement 1D-E*). Together, these data indicate that lobe cannibalism and autophagy do not subtract the number of PGC mtDNAs to a defined number, but rather divide the population of PGC mtDNAs present by a fixed proportion.

## Lobe cannibalism and autophagy generate an mtDNA low point and set point in germline stem cells

Our results so far suggest that lobe cannibalism and autophagy could contribute to a germline mtDNA bottleneck by halving the number of PGC mtDNAs. However, if the initial cycles of larval germline proliferation proceed in the absence of bulk mtDNA replication, the number of mtDNAs per germ cell would continue to drop and an mtDNA low point (per cell) would occur at a later stage of germline development. When L1 larvae first encounter food, PGCs exit from quiescence and begin to proliferate, forming a population of undifferentiated germline stem cells (GSCs) (*Fukuyama et al., 2006*; *Hubbard and Schedl, 2019*). It is not known whether germline mtDNA replication has begun at this stage. Previous quantitative PCR (qPCR) experiments on whole worms first revealed a significant expansion of germline mtDNAs after the L3 larval stage (*Bratic et al., 2009*; *Tsang and Lemire, 2002a*). However, these experiments might have lacked the resolution to detect an increase in mtDNAs were it to occur within the relatively small number of GSCs present in whole L1 larvae.

To determine if mtDNAs replicate as L1 PGCs exit quiescence and divide to produce GSCs, we quantified TFAM foci as PGCs in fed L1s began to proliferate as GSCs. To circumvent the mtDNA replication defects that we noted in *TFAM-GFP* worms, we utilized split GFP, a form of bimolecular fluorescence complementation that brings together the $GFP_{1-10}$ and $GFP_{11}$ fragments of GFP, which are non-fluorescent until reunited (*Cabantous et al., 2005*; *Kamiyama et al., 2016*); this approach allowed us to tag *TFAM* (*hmg-5*) endogenously with the much smaller $GFP_{11}$ tag. To visualize TFAM-$GFP_{11}$, we expressed a mitochondrial matrix-targeted, PGC-specific, $GFP_{1-10}$ [Mito-$GFP_{1-10}^{(PGC)}$]. The $GFP_{1-10}$ alone was minimally fluorescent, but upon binding to $GFP_{11}$ formed a functional fluorophore (*Figure 3—figure supplement 1A-B*). Within PGCs, TFAM-$GFP_{11}$ detected with Mito-$GFP_{1-10}^{(PGC)}$

showed an identical localization pattern to TFAM-GFP (*Figure 3A–B*), and did not cause significant defects in mtDNA copy number (*Figure 3—figure supplement 1C*). Larvae fed beginning at the L1 stage showed a progressive increase in the number of TFAM-GFP$_{11}$ foci per germ line (*Figure 3C–F*, compare with *Figure 3B*). The TFAM-GFP$_{11}$ foci numbers began to increase even before the first division of the PGCs was complete (early-L1, *Figure 3C and G*), and continued to expand through the L2 stage (an average of 22 GSCs), when we stopped our analysis (*Figure 3D–G*). In contrast to the increasing number of TFAM-GFP$_{11}$ foci per *germ line* over this period, the number of foci per *germ cell* remained constant after a transient spike in early-L1s (two cells), and stabilized at a number of foci similar to that of L1 that had not been fed (*Figure 3H*).

To complement these experiments, we sorted GSCs (*Figure 3—figure supplement 2*) from fed mid-L1 larvae (containing an average of 4 GSCs) and L2 larvae (containing an average of 18 GSCs), and counted the number of mtDNA molecules per GSC. Consistent with our TFAM-GFP$_{11}$ observations, the number of mtDNAs per germ line increased over this period (*Figure 3I*), although the number of mtDNAs per GSC remained constant and similar to that in starved L1s (~200; *Figure 3J*). Together, these results indicate that mtDNAs replicate in bulk as L1 PGCs begin to divide to form GSCs, and thereafter balance mtDNA replication with cell division to maintain a constant number of mtDNAs per GSC, through at least the L2 stage.

The observation that GSCs contain the same number of mtDNAs as L1 PGCs suggests that lobe cannibalism and autophagy might function to reduce PGC mtDNA numbers to an optimal level. To explore this hypothesis, we examined mtDNA number in GSCs of *nop-1* mutants, since we found that *nop-1* L1 PGCs contain excess mtDNAs. Remarkably, *nop-1* GSCs isolated from mid-L1 and -L2 larvae contained a similar number of mtDNAs as did wild-type GSCs (~200; *Figure 3K–L*). These results suggest that GSCs actively coordinate mtDNA replication with cell division to maintain ~200 mtDNA per cell, even if excess mtDNAs are present at the onset of germline expansion.

## Purifying selection reduces mutant mtDNA heteroplasmy in PGCs independently of lobe cannibalism

Our experiments so far have not addressed whether PGCs eliminate mitochondria indiscriminately, or alternatively, if poorly functioning mitochondria, containing high levels of mutant mtDNA, are preferentially targeted for destruction. To examine this question, we investigated PGCs containing the *uaDf5* mtDNA deletion. The *uaDf5* deletion removes 3.1 kb of the mitochondrial genome, including several essential genes (*Figure 4A*), and therefore can exist only when in heteroplasmy with wild-type mtDNA (*Tsang and Lemire, 2002b*). However, *uaDf5* persists stably because it is preferentially replicated compared with wild-type mtDNA (*Yang et al., 2022*; *Gitschlag et al., 2016*; *Gitschlag et al., 2020*; *Lin et al., 2016*). Our experiments above suggest that bulk mtDNA expansion does not occur until PGCs differentiate and divide in L1 larvae, potentially providing an opportunity for purifying selection to reduce *uaDf5* levels before larval germline growth begins.

First, we measured *uaDf5* heteroplasmy in whole embryos, embryonic PGCs, and L1 PGCs, as well as in GSCs of fed larvae (*Figure 4B*, *Figure 4—figure supplement 1A-C*). The *uaDf5* deletion occurred at 48% ± 0.3 heteroplasmy in embryonic PGCs, which was nearly identical to its heteroplasmy in whole embryos (*Figure 4B*), suggesting that there is no strong selection against *uaDf5* during embryogenesis prior to PGC birth. However, in L1 PGCs, *uaDf5* heteroplasmy dropped by 4.5% (*Figure 4B*). This effect was not specific to *uaDf5*, as PGC heteroplasmy of the 1.5 kb *mptDf2* deletion was also reduced between embryogenesis and the L1 stage (*Figure 4—figure supplement 1F*). Within starved L1 PGCs, *uaDf5* was present at 43.5% ± 0.5 heteroplasmy and was maintained at a similar level after the PGCs divided once to form four GSCs (mid-L1 stage). However, by the L2 stage (average of 20 GSCs), *uaDf5* heteroplasmy increased to 53% ± 1.7 a level nearly identical to that of whole adult worms (*Figure 4B*). These findings suggest that PGCs utilize purifying selection to reduce levels of mutant mtDNAs at a stage when *uaDf5* cannot take advantage of bulk mtDNA replication to expand selfishly within the germ line. However, once the number of mtDNAs expands in larval GSCs, the percentage of *uaDf5* mutant mtDNAs can once again rise.

To test whether lobe cannibalism is responsible for purifying selection against *uaDf5* in PGCs, we examined *uaDf5* heteroplasmy in *nop-1* mutant PGCs. Similar to wild type, *uaDf5* PGCs reduced their total mtDNA content ~twofold between embryogenesis and L1, and as expected, *nop-1; uaDf5* PGCs failed to reduce their mtDNA significantly (*Figure 4C*). Surprisingly, we found that in *nop-1;*

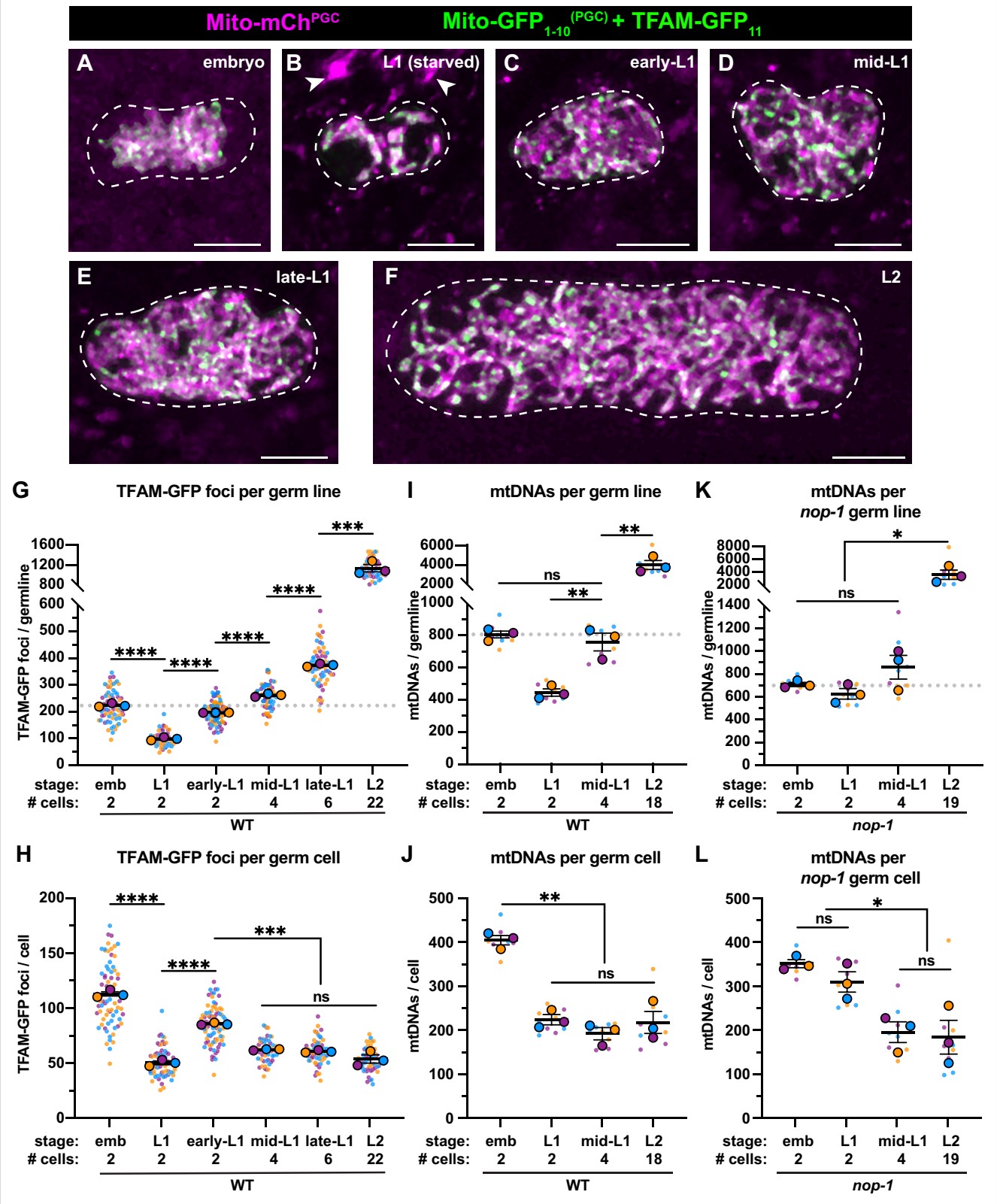

**Figure 3.** Primordial germ cell (PGC) lobe cannibalism and autophagy generate a mitochondrial DNA (mtDNA) low point and set point. (**A–F**) Germline mitochondria and mitochondrial transcription factor-A (TFAM)-GFP$_{11}$ in live embryos and larvae at the indicated stage. Dashed lines outline the PGCs or germline stem cells (GSCs). (**G–H**) Quantification of TFAM-GFP$_{11}$ foci per germ line (**G**) and per germ cell (**H**) in embryos and larvae. (**I–J**) Quantification of mtDNAs per germ line (**I**) or per germ cell (**J**) in embryos and larvae; data shown for PGC mtDNA copy number in embryos and starved L1s are

*Figure 3 continued on next page*

*Figure 3 continued*

provided for comparison and originate from *Figure 2B*. (**K–L**) Quantification of mtDNAs per germ line (**K**) or per germ cell (**L**) in *nop-1* mutant embryos and larvae; data shown for PGC mtDNA copy number in *nop-1* mutant embryos and starved L1s are provided for comparison and originate from *Figure 2E*. Data in graphs: small dots are individual animals (TFAM-GFP$_{11}$ measurements) or technical replicates (droplet digital PCR [ddPCR] experiments) from three color-coded biological replicates; the mean from each experiment is shown as a larger circle, the mean of means as a horizontal line, and the SEM as error bars. n.s., not significant (p>0.05), *p≤0.05, **p≤0.01, ***p≤0.001, ****p≤0.0001 unpaired two-tailed Student's *t*-test. Scale bars, 5 µm.

The online version of this article includes the following source data and figure supplement(s) for figure 3:

**Source data 1.** Related to *Figure 3G–L*.

**Figure supplement 1.** TFAM-GFP$_{11}$ visualization and effect on mitochondrial DNA (mtDNA) copy number.

**Figure supplement 1—source data 1.** Related to *Figure 3—figure supplement 1C*.

**Figure supplement 2.** Fluorescence activated cell sorting (FACS), ploidy, and purity of sorted larval germline stem cells (GSCs).

**Figure supplement 2—source data 1.** Related to *Figure 3—figure supplement 2E*.

---

*uaDf5* mutants, *uaDf5* heteroplasmy still decreased from embryonic PGCs to L1 PGCs (*Figure 4D–E*). We conclude that lobe cannibalism is not responsible for the reduction in *uaDf5* heteroplasmy within PGCs, implicating an alternative pathway in PGC mtDNA purifying selection.

## Autophagy eliminates a subset of PGC mitochondria non-selectively

Several adaptor proteins function upstream of the autophagy pathway to specifically eliminate mitochondria – a process called mitophagy. The mitophagy receptor BCL2 interacting protein 3 like (BNIP3L) is required for autophagy-driven mitochondrial clearance in erythrocytes (*Meiklejohn et al., 2007*; *Sandoval et al., 2008*; *Schweers et al., 2007*; *Zhang et al., 2009*) and for mtDNA purifying selection in the *Drosophila* ovary (*Lieber et al., 2019*). To test whether mitophagy or autophagy preferentially removes *uaDf5* mtDNAs in PGCs, we sorted embryonic and L1 PGCs in *uaDf5* mutants containing putative null mutations in *dct-1/BNIP3L,* as well as in *atg-18* and *atg-13*, which block autophagy at the elongation and initiation steps, respectively (*Palmisano and Meléndez, 2019*; *Tian et al., 2009*). Surprisingly, we found no defect in either PGC mtDNA copy number reduction or purifying selection in *dct-1/BNIP3L; uaDf5* mutants (*Figure 5—figure supplement 1*). The L1 PGCs in *atg-18* and *atg-13* mutants had reduced numbers of total mtDNAs compared with embryonic PGCs, although a smaller percentage (*atg-18*: 25% and *atg-13*: 20%) of mtDNAs were eliminated compared to *uaDf5* alone (52%; *Figure 5A–B*). Consistent with our findings above (see *Figure 2*), these data suggest that the autophagy pathway, but not the mitophagy receptor DCT-1/BNIP3L, acts in parallel with lobe cannibalism and is partially responsible for the reduction of mtDNAs in PGCs. Unexpectedly, in both *atg-13; uaDf5* and *atg-18; uaDf5* mutants, *uaDf5* heteroplasmy was still reduced in L1 PGCs compared to embryonic PGCs (*Figure 5C–D*). We conclude that autophagy likely eliminates a subset of mitochondria and mtDNAs within PGCs non-selectively, but is not responsible for purifying selection against *uaDf5* mutant mtDNA. Consistent with this interpretation, we observed acidified mitochondria [mCherry(+) Dendra(-)] in *uaDf5* PGCs at comparable frequencies to wild-type PGCs (*Figure 5E and G*, *Figure 2—figure supplement 4*), and acidified foci were completely absent in *atg-18; uaDf5* null mutant embryos (*Figure 5F–G*).

## PINK1 mediates autophagy-independent mtDNA purifying selection in PGCs

The PINK1/Parkin signaling pathway, which consists of the mitochondrial kinase PINK1 and its effector ubiquitin ligase Parkin, can recognize and mark defective mitochondria for destruction either via autophagy or through autophagy-independent pathways (*Zhang et al., 2019*; *Hammerling et al., 2017*; *McLelland et al., 2014*). *C. elegans* contains single orthologs of PINK1 (encoded by *pink-1*) and Parkin (encoded by *pdr-1*) (*Hamamichi et al., 2008*; *Springer et al., 2005*). To address whether PINK1 or Parkin are required for purifying selection of *uaDf5*, we examined *uaDf5* heteroplasmy in PGCs with putative null mutations in *pink-1*, *pdr-1*, and *pink-1; pdr-1* double mutants. As expected, single and double mutants had reduced mtDNA content in L1 PGCs compared to embryonic PGCs, although to a lesser extent than *uaDf5* controls (*Figure 6A–B*). However, even though *uaDf5* heteroplasmy was markedly higher in all three backgrounds compared to *uaDf5* controls (see Discussion), only *pink-1,*

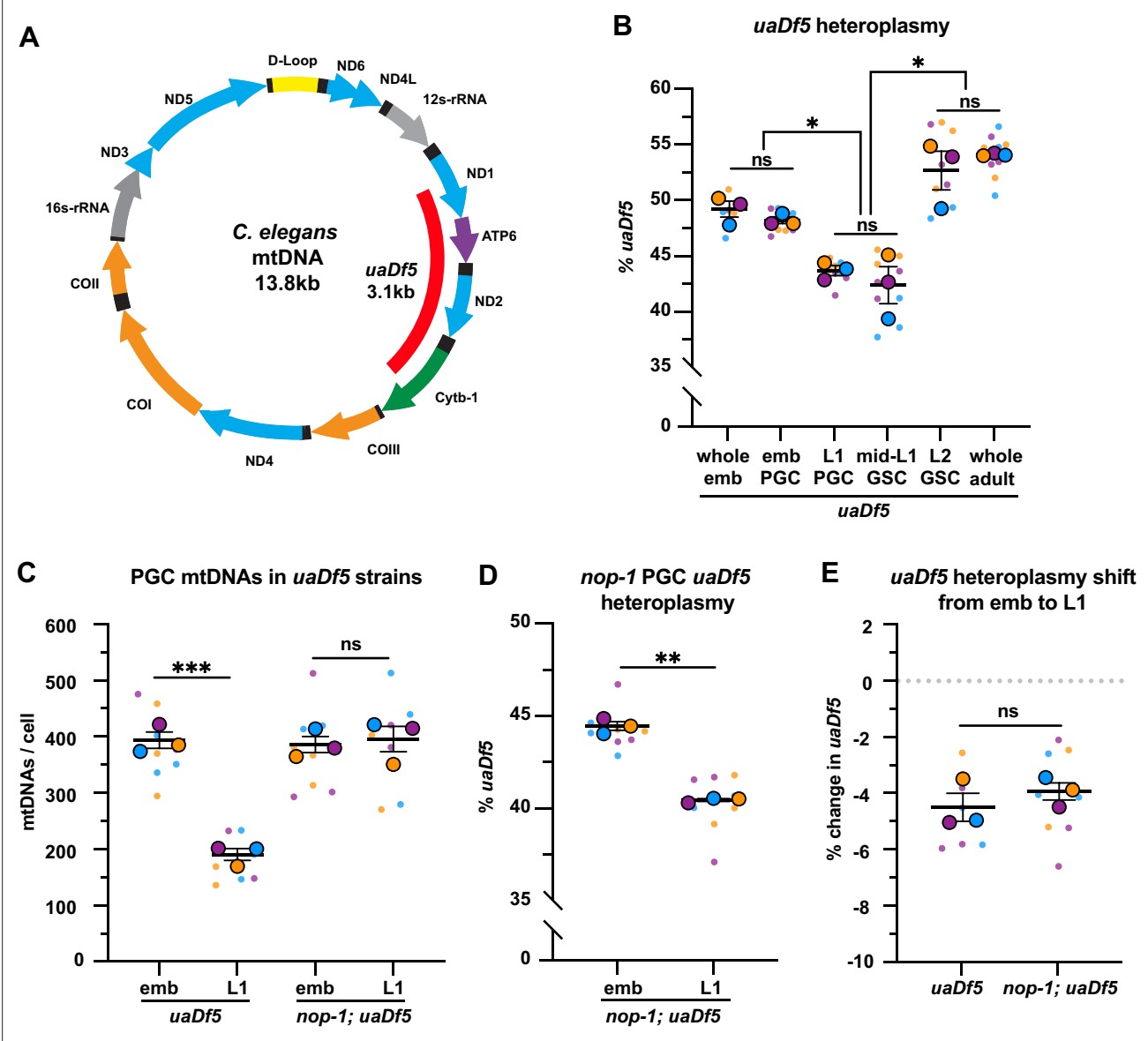

**Figure 4.** Primordial germ cells (PGCs) reduce *uaDf5* heteroplasmy independently of lobe cannibalism. (**A**) Schematic of *C. elegans* mitochondrial DNA (mtDNA); genes are indicated with colored arrows and the region deleted in *uaDf5* is shown with a red bar. (**B**) Quantification of *uaDf5* heteroplasmy in whole embryos, sorted PGCs or germline stem cells (GSCs), or whole adults at the indicated stages. (**C**) Quantification of mtDNA copy number in PGCs of *uaDf5* and *nop-1; uaDf5* mutants. (**D**) Quantification of *uaDf5* heteroplasmy in *nop-1; uaDf5* mutant PGCs. (**E**) Data from (**B and D**) presented as change in heteroplasmy shift from embryonic to L1 PGCs. Data in graphs: small dots are three technical replicates of droplet digital PCR (ddPCR) quantification from each of three color-coded biological replicates; the technical replicate mean from each experiment is shown as a larger circle, the mean of means as a horizontal line, and the SEM as error bars. n.s., not significant (p>0.05), *p≤0.05, **p≤0.01, ***p≤0.001, paired (**B, D**) and unpaired (**B, C, E**) two-tailed Student's *t*-test.

The online version of this article includes the following source data and figure supplement(s) for figure 4:

**Source data 1.** Related to *Figure 4B–E*.

**Figure supplement 1.** Droplet digital PCR (ddPCR) primers, detection of mitochondrial DNA (mtDNA) deletions, and *mptDf2* inheritance in primordial germ cells (PGCs).

**Figure supplement 1—source data 1.** Related to *Figure 4—figure supplement 1D-F*.

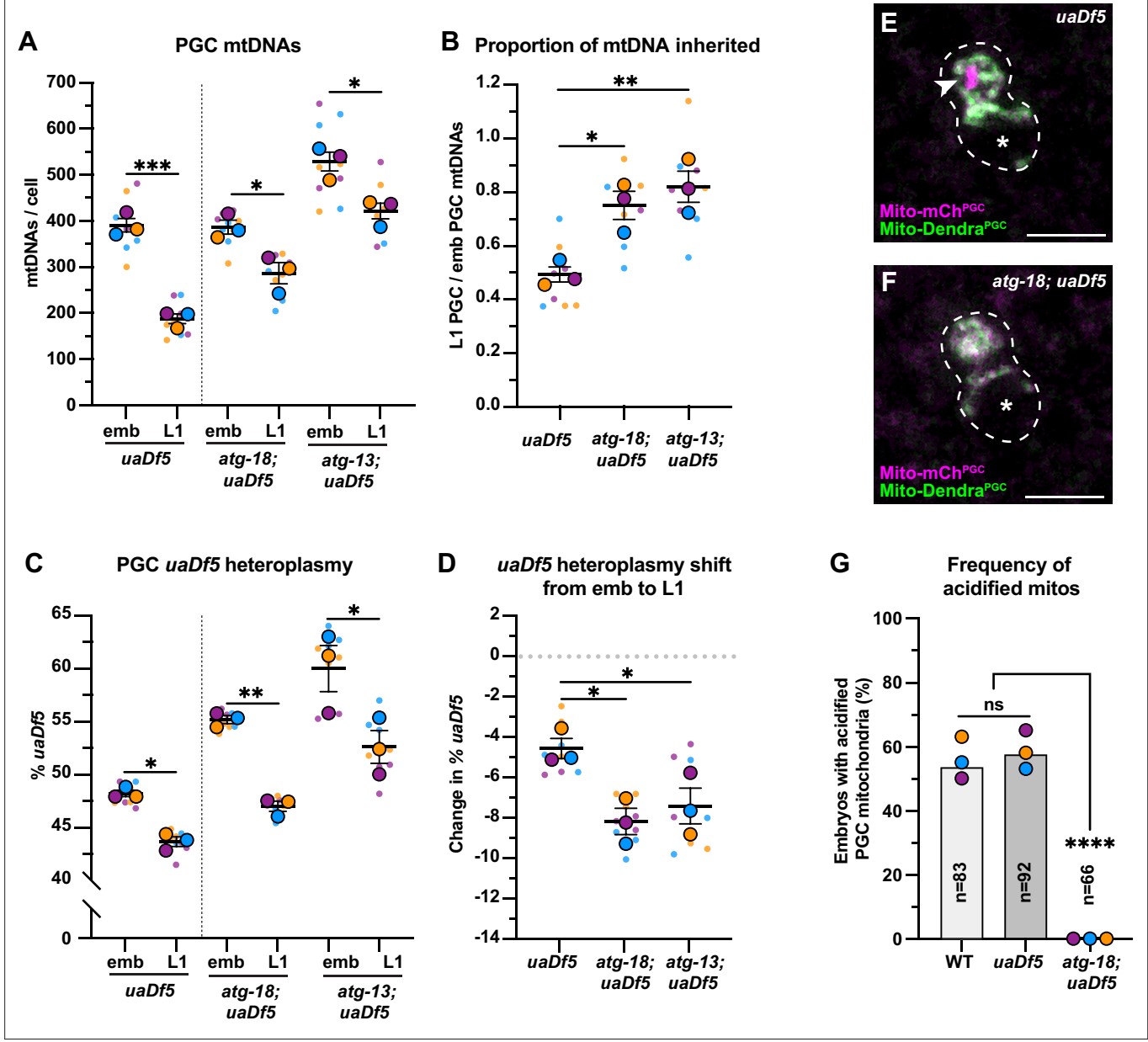

**Figure 5.** Autophagy eliminates a pool of primordial germ cell (PGC) mitochondrial DNAs (mtDNAs) non-selectively. (**A**) mtDNA copy number in *atg-18; uaDf5*, and *atg-13; uaDf5* embryonic and L1 PGCs; data shown for *uaDf5* are provided for comparison, originate from *Figure 4C*, and are delineated with a dashed line. (**B**) Data from (**A**) presented as proportion of embryonic PGC mtDNAs inherited by L1 PGCs. (**C**) *uaDf5* heteroplasmy in *atg-18; uaDf5* and *atg-13; uaDf5* PGCs; data shown for *uaDf5* are provided for comparison, originate from *Figure 4B*, and are delineated with a dashed line. (**D**) Data from (**C**) presented as change in heteroplasmy shift from embryonic to L1 PGCs. Data in graphs: small dots are three technical replicates of droplet digital PCR (ddPCR) quantification from each of three color-coded biological replicates; the technical replicate mean from each experiment is shown as a larger circle, the mean of means as a horizontal line, and the SEM as error bars. n.s., not significant (p>0.05), *p≤0.05, **p≤0.01, ***p≤0.001 paired (**C**) and unpaired (**A, B, D**) two-tailed Student's *t*-test. (**E–F**) Acidified mitochondria (magenta regions, arrowhead in E) in *uaDf5* PGCs (**E**) and absent in *atg-18; uaDf5* PGCs (**F**). (**G**) Percentage of embryos with acidified mitochondria in PGCs. Three biological replicates (N≥16) are shown as colored circles, with peak a of the bar on the graph representing the mean. Fisher's exact test was used to determine statistical significance. n.s., not significant (p>0.05), ****p≤0.0001.

The online version of this article includes the following source data and figure supplement(s) for figure 5:

**Source data 1.** Related to *Figure 5A–D and G*.

**Figure supplement 1.** *dct-1/BNIP3L* is not required for mitochondrial DNA (mtDNA) regulation in primordial germ cells (PGCs).

**Figure supplement 1—source data 1.** Related to *Figure 5—figure supplement 1A-C*.

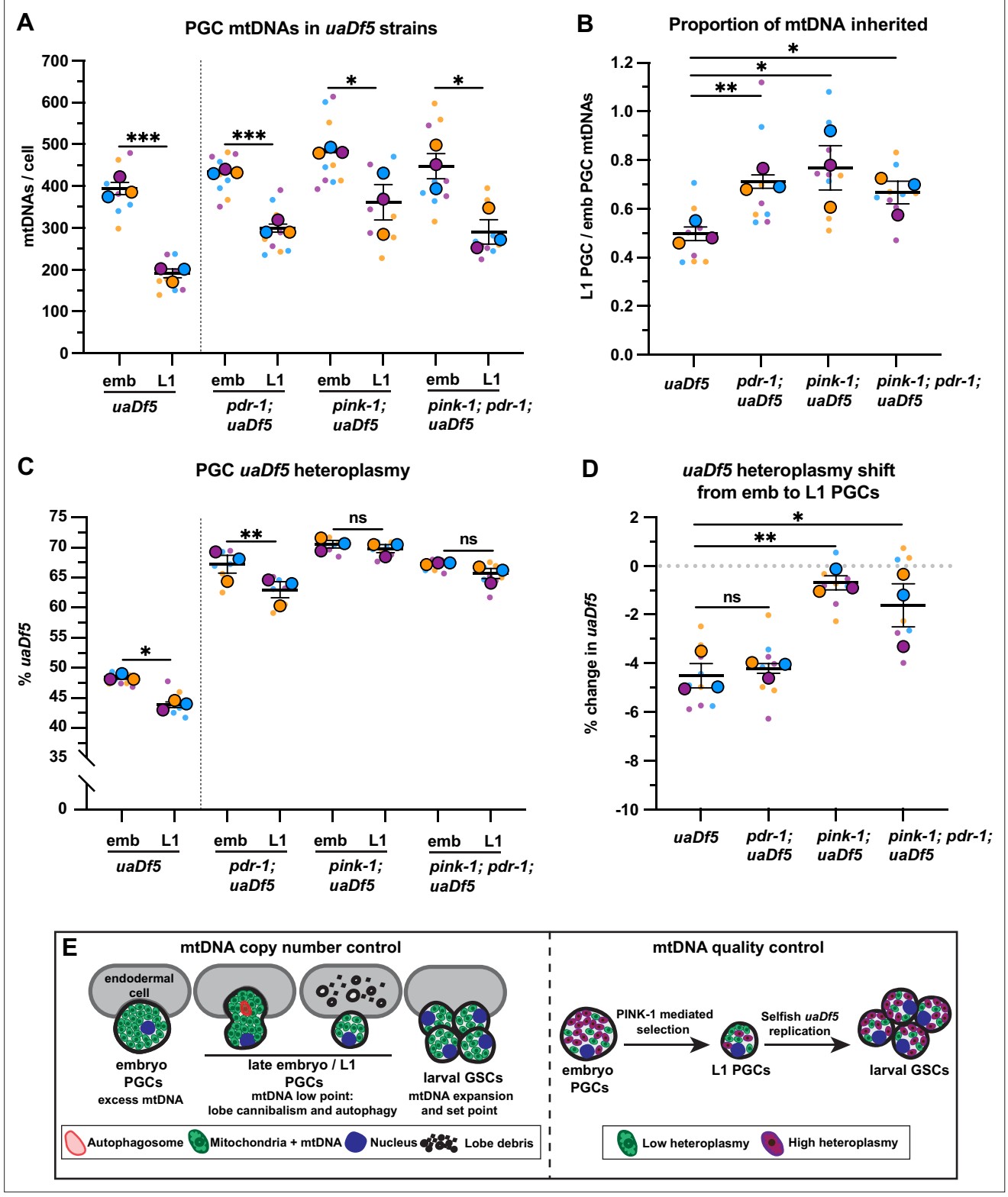

**Figure 6.** PINK-1 mediates mitochondrial DNA (mtDNA) purifying selection in primordial germ cells (PGCs). (a) mtDNA copy number in *pdr-1; uaDf5*, *pink-1; uaDf5*, and *pink-1; pdr-1; uaDf5* embryonic and L1 PGCs; data shown for *uaDf5* are provided for comparison, originate from **Figure 4C**, and are delineated with a dashed line. (**B**) Data from (**A**) presented as proportion of embryonic PGC mtDNAs inherited by L1 PGCs. (**C**) Percent *uaDf5* heteroplasmy in *pdr-1; uaDf5*, *pink-1; uaDf5*, and *pink-1; pdr-1; uaDf5* PGCs; data shown for *uaDf5* are provided for comparison, originate from **Figure 4B**, and are delineated with a dashed line. (**D**) Data from (**C**) presented as change in heteroplasmy shift from embryonic to L1 PGCs. Data in

*Figure 6 continued on next page*

*Figure 6 continued*

graphs: small dots are three technical replicates of droplet digital PCR (ddPCR) quantification from each of three color-coded biological replicates; the technical replicate mean from each experiment is shown as a larger circle, the mean of means as a horizontal line, and the SEM as error bars. n.s., not significant (p>0.05), *p≤0.05, **p≤0.01, ***p≤0.001, paired (**C**) and unpaired (**A, B, D**) two-tailed Student's *t*-test. (**E**) Model for regulation of mtDNA quantity and quality in PGCs and germline stem cells (GSCs).

The online version of this article includes the following source data for figure 6:

**Source data 1.** Related to *Figure 6A–D*.

and *pink-1; pdr-1* double mutants, but not *pdr-1* single mutants, abrogated the reduction in *uaDf5* heteroplasmy between embryonic and L1 PGCs (*Figure 6C–D*). Taken together, these findings indicate that PINK-1 alone, acting independently of PDR-1/Parkin, is required for autophagy-independent purifying selection against mutant mtDNAs within *C. elegans* PGCs.

## Discussion

Our findings show that *C. elegans* PGCs actively regulate both mtDNA quantity and quality, but do so through independent and parallel mechanisms (*Figure 6E*). The cannibalism of PGC lobes, combined with non-selective general autophagy, generates a germ cell mtDNA low point and set point, whereas PINK-1 selectively reduces mutant mtDNA heteroplasmy. We propose that this combined regulation optimizes the founding population of mitochondria before the germ line expands and differentiates in larvae.

What is the purpose of reducing the number of mtDNAs in PGCs? We postulate that embryonic PGCs inherit excess maternal mtDNAs, as they are born from relatively few embryonic cell divisions (*Sulston et al., 1983*), and together lobe cannibalism and autophagy halve PGC mtDNA copy number to establish a level that is maintained in GSCs as bulk mtDNA replication ensues. Having ~200 mtDNAs per PGC appears to be important, as even when L1 PGCs inherit an excess of mtDNAs in *nop-1* mutants, mtDNA copy number quickly resets to ~200 shortly after PGCs differentiate into proliferating GSCs. These findings indicate that GSCs balance mtDNA replication with cell division to reach an mtDNA set point of ~200 that is actively maintained. It is possible that this number of mtDNAs is optimal for balancing mitochondrial function with germ cell size and physiology. Whether attaining ~200 mtDNA per PGC functions as a genetic bottleneck remains unknown. However, it is worth noting that several studies have detected a comparable number of mtDNAs in early mouse and zebrafish PGCs (*Jenuth et al., 1996*; *Otten et al., 2016*; *Wai et al., 2008*), and that this stage has been proposed as an mtDNA genetic bottleneck in mammals based on simulation studies (*Cree et al., 2008*).

Whereas PGC lobe cannibalism and autophagy produce a stochastic reduction in mtDNA number, we found that PINK-1 specifically reduces the fraction of mutant mtDNAs in PGCs. While the effect of PINK-1-mediated selection against *uaDf5* is modest, even small decreases in heteroplasmy could have important evolutionary consequences. For example, individual selection events against de novo mtDNA mutations could eliminate them from the germ line permanently. In other systems, PINK-1 can eliminate poorly functioning mitochondria by recruiting Parkin and inducing mitophagy (*Palikaras et al., 2018*). However, we find no role for Parkin or autophagy in PGC mtDNA purifying selection, although autophagy is partially required for reducing PGC mtDNA number. It is possible that autophagy serves a separate quality control function in PGCs, perhaps by removing mitochondria with high levels of oxidative stress, as previous work suggests that PGC mitochondria are highly oxidized compared to those in somatic cells (*Abdu et al., 2016*).

Alternative mechanisms of PINK1-mediated mitochondrial elimination have been described in cultured mammalian cells, such as direct targeting of endolysosomes, formation of mitochondria-derived vesicles, and extracellular secretory release (*Hammerling et al., 2017*; *McLelland et al., 2014*; *Tan et al., 2022*). It will be important in future studies to determine whether PINK-1 operates in PGCs through one of these pathways or via a novel mechanism. It is worth noting that *uaDf5* heteroplasmy in embryonic PGCs is higher in *pink-1*, *pdr-1*, and autophagy mutants, suggesting that these pathways have roles in purifying selection during other stages of germ line development, as they do in somatic cells (*Ahier et al., 2021*; *Bess et al., 2012*; *Kandul et al., 2016*).

Purifying selection in *C. elegans* PGCs differs from mechanisms described in the *Drosophila* ovary, where mutant mtDNAs are eliminated through mitochondrial fission followed by BNIP3L-mediated autophagy, and mutant mtDNA replication is selectively inhibited by PINK1 (*Chen et al., 2020*; *Lieber et al., 2019*; *Zhang et al., 2019*; *Hill et al., 2014*; *Ma et al., 2014*). We cannot exclude the possibility that low levels of mtDNA replication occur in *C. elegans* PGCs, though we do not observe robust mtDNA expansion until PGCs differentiate into GSCs in fed L1 larvae. This finding is also supported by our observation that *uaDf5* heteroplasmy decreases in PGCs, whereas *uaDf5* is known to selfishly expand through preferential mtDNA replication (*Yang et al., 2022*). Indeed, we showed that as PGCs differentiate to GSCs, *uaDf5* heteroplasmy rapidly increases to levels found in the adult. Previous work has suggested that selection also occurs during *C. elegans* oogenesis, although the mechanism is unknown (*Gitschlag et al., 2020*; *Ahier et al., 2018*). It will be interesting to determine if these different means of achieving purifying selection are stage-specific (ovary versus PGC), or reveal that multiple mechanisms can be used toward the common goal of eliminating mutant mtDNA genomes from the germ line.

# Methods

## Key resources table

| Reagent type (species) or resource | Designation | Source or reference | Identifiers | Additional information |
|---|---|---|---|---|
| Strain, strain background (*C. elegans*) | *C. elegans* wild isolate | *Caenorhabditis* Genetics Center (CGC) | N2 | |
| Strain, strain background (*C. elegans*) | *atg-18(gk378)* V | CGC | VC893 | |
| Strain, strain background (*C. elegans*) | *atg-13(bp414)* III | CGC | HZ1688 | |
| Strain, strain background (*C. elegans*) | *pdr-1(gk448)* III | CGC | VC1024 | |
| Strain, strain background (*C. elegans*) | *xnSi1 [mex-5p::GFP-PH::nos-2 3'UTR, unc-119(+)]* II; *unc-119(ed3)* III | *Chihara and Nance, 2012* | FT563 | |
| Strain, strain background (*C. elegans*) | *hmg-5(xn107[hmg-5-GFP])* IV | This study | FT2064 | *hmg-5(xn107)* made by Clustered Regularly Interspaced Short Palindromic Repeats (CRISPR). |
| Strain, strain background (*C. elegans*) | *hmg-5(xn107[hmg-5-GFP])* IV; *xnIs360 [pMRR08(mex-5p::mCherry-PH::nos-2 3'UTR, unc-119(+))]* V | This study | FT2133 | Shown in *Figure 1J–K*. |
| Strain, strain background (*C. elegans*) | *xnSi67 [pYA57(mex-5p::mito-tomm-20^{1-54}-Dendra2::nos-2 3'UTR, unc-119(+))]* I; *unc-119(ed3)* III | This study | FT1885 | Made by Mos1-mediated single copy insertion (MosSCI). |
| Strain, strain background (*C. elegans*) | *xnSi67 [pYA57(mex-5p::mito-tomm-20^{1-54}-Dendra2::nos-2 3'UTR, unc-119(+))]* I; *xnIs360 [pMRR08(mex-5p::mCherry-PH::nos-2 3'UTR, unc-119(+))]* V | This study | FT1900 | Shown in *Figure 1B–D*, *Figure 1—figure supplement 1*, and *Figure 2C–D*. |
| Strain, strain background (*C. elegans*) | *xnSi67 [pYA57(mex-5p::mito-tomm-20^{1-54}-Dendra2::nos-2 3'UTR, unc-119(+))]* I; *xnSi45 [pYA11(mex-5p::mCherry-moma-1::nos-2 3'UTR, unc-119(+))]* II | This study | FT2366 | Shown in *Figure 1G* and *Figure 2—figure supplement 4*. |
| Strain, strain background (*C. elegans*) | *xnSi67 [pYA57(mex-5p::mito-tomm-20^{1-54}-Dendra2::nos-2 3'UTR, unc-119(+))]* I; *xnSi45 [pYA11(mex-5p::mCherry-moma-1::nos-2 3'UTR, unc-119(+))]* II; *uaDf5 /+* mtDNA | This study | FT2414 | Shown in *Figure 5E*. |
| Strain, strain background (*C. elegans*) | *xnSi67 [pYA57(mex-5p::mito-tomm-20^{1-54}-Dendra2::nos-2 3'UTR, unc-119(+))]* I; *xnSi45 [pYA11(mex-5p::mCherry-moma-1::nos-2 3'UTR, unc-119(+))]* II; *atg-18(gk378)* V; *uaDf5 /+* mtDNA | This study | FT2417 | Shown in *Figure 5F*. |

*Continued on next page*

*Continued*

| Reagent type (species) or resource | Designation | Source or reference | Identifiers | Additional information |
|---|---|---|---|---|
| Strain, strain background (*C. elegans*) | *xnSi73 [mex-5p::GFP$_{1-10}$::nos-2 3'UTR, unc-119(+)] I; xnSi45 [pYA11(mex-5p::mCherry-moma-1::nos-2 3'UTR)] II* | This study | FT2128 | *xnSi73* made by CRISPR, see Methods. |
| Strain, strain background (*C. elegans*) | *xnSi85 [mex-5p::mito(matrix)-GFP$_{1-10}$::nos-2 3'UTR] I; xnSi45 [pYA11(mex-5p::mCherry-moma-1::nos-2 3'UTR, unc-119(+))] II* | This study | FT2293 | *xnSi85* made by CRISPR, see Methods |
| Strain, strain background (*C. elegans*) | *xnSi85 [mex-5p::mito(matrix)-GFP$_{1-10}$::nos-2 3'UTR] I; xnSi45 [pYA11(mex-5p::mCherry-moma-1::nos-2 3'UTR, unc-119(+))] II; hmg-5(xn168[hmg-5-GFP$_{11}$]) IV* | This study | FT2296 | *hmg-5(xn168)* made by CRISPR. Shown in **Figure 3A–F**, and **Figure 3—figure supplement 1A, B** |
| Strain, strain background (*C. elegans*) | *glh-1(sam24[glh-1-GFP-3xFLAG]) I; xnIs510 [pYA12(ehn-3p::mCherry-PH, unc-119(+))] II* | This study | FT2279 | *glh-1(sam24)* a gift from Dustin Updike (MDI Biological Laboratory) (**Marnik et al., 2019**). Base strain used for all cell sorting. Related to **Figures 2–6**. |
| Strain, strain background (*C. elegans*) | *glh-1(sam24[glh-1-GFP-3xFLAG]) I; xnIs510 [pYA12(ehn-3p::mCherry-PH, unc-119(+))] II; uaDf5 /+ mtDNA* | This study | FT2283 | Related to data shown in **Figure 4** |
| Strain, strain background (*C. elegans*) | *glh-1(sam24[glh-1-GFP-3xFLAG]) I; xnIs510 [pYA12(ehn-3p::mCherry-PH, unc-119(+))] II; hmg-5(xn107[hmg-5-GFP]) IV* | This study | FT2312 | Related to data shown in **Figure 2**. |
| Strain, strain background (*C. elegans*) | *glh-1(sam24[glh-1-GFP-3xFLAG]) I; xnIs510 [pYA12(ehn-3p::mCherry-PH, unc-119(+))] II; nop-1(full CRISPR deletion) III* | This study | FT2323 | *nop-1* deletion a gift from Heng-Chi Lee (University of Chicago) (**Zhang et al., 2018**). Related to data shown in **Figures 2–3**. |
| Strain, strain background (*C. elegans*) | *glh-1(sam24[glh-1-GFP-3xFLAG]) I; xnIs510 [pYA12(ehn-3p::mCherry-PH, unc-119(+))] II; nop-1(full CRISPR deletion) III; uaDf5 /+ mtDNA* | This study | FT2332 | Related to data shown in **Figure 4C–E** |
| Strain, strain background (*C. elegans*) | *glh-1(sam24[glh-1-GFP-3xFLAG]) I; xnIs510 [pYA12(ehn-3p::mCherry-PH, unc-119(+))] II; atg-18(gk378) V; uaDf5 /+ mtDNA* | This study | FT2347 | Related to data shown in **Figure 5** |
| Strain, strain background (*C. elegans*) | *glh-1(sam24[glh-1-GFP-3xFLAG]) I; xnIs510 [pYA12(ehn-3p::mCherry-PH, unc-119(+))] II; atg-13(bp414) III; uaDf5 /+ mtDNA* | This study | FT2402 | Related to data shown in **Figure 5** |
| Strain, strain background (*C. elegans*) | *glh-1(sam24[glh-1-GFP-3xFLAG]) I; xnIs510 [pYA12(ehn-3p::mCherry-PH, unc-119(+))] II; nop-1(full CRISPR deletion) III; atg-18(gk378) V* | This study | FT2443 | Related to data shown in **Figure 2** |
| Strain, strain background (*C. elegans*) | *glh-1(sam24[glh-1-GFP-3xFLAG]) I; xnIs510 [pYA12(ehn-3p::mCherry-PH, unc-119(+))] II; pdr-1(gk448) III; uaDf5 /+ mtDNA* | This study | FT2364 | Related to data shown in **Figure 6** |
| Strain, strain background (*C. elegans*) | *glh-1(sam24[glh-1-GFP-3xFLAG]) I; pink-1(xn199[pink-1(STOP-IN)]); xnIs510 [pYA12(ehn-3p::mCherry-PH, unc-119(+))] II; uaDf5 /+ mtDNA* | This study | FT2432 | *pink-1(xn199)* made by CRISPR. Related to data shown in **Figure 6**. |
| Strain, strain background (*C. elegans*) | *glh-1(sam24[glh-1-GFP-3xFLAG]) I; pink-1(xn199[pink-1(STOP-IN)]); xnIs510 [pYA12(ehn-3p::mCherry-PH, unc-119(+))] II; pdr-1(gk448) III; uaDf5 /+ mtDNA* | This study | FT2378 | Related to data shown in **Figure 6**. |
| Strain, strain background (*C. elegans*) | *glh-1(sam24[glh-1-GFP-3xFLAG]) I; xnIs510 [pYA12(ehn-3p::mCherry-PH, unc-119(+))] II; mptDf2 /+ mtDNA* | This study | FT2387 | Related to data shown in **Figure 4—figure supplement 1**. |
| Strain, strain background (*C. elegans*) | *glh-1(sam24[glh-1-GFP-3xFLAG]) I; xnIs510 [pYA12(ehn-3p::mCherry-PH, unc-119(+))] II; dct-1(xn192[dct-1(STOP-IN)]) X uaDf5 /+mtDNA* | This study | FT2339 | *dct-1(xn192)* made by CRISPR. Related to data shown in **Figure 5—figure supplement 1**. |
| Strain, strain background (*C. elegans*) | *xnSi67 [pYA57(mex-5p::mito(tomm-20$^{1-54}$)-Dendra2::nos-2 3'UTR)] I; xnSi45 [pYA11(mex-5p::mCherry-moma-1::nos-2 3'UTR, unc-119(+))] II; uaDf5 /+ mtDNA* | This study | FT2414 | Shown in **Figure 5E**. |

*Continued on next page*

*Continued*

| Reagent type (species) or resource | Designation | Source or reference | Identifiers | Additional information |
|---|---|---|---|---|
| Strain, strain background (*C. elegans*) | *xnSi67 [pYA57(mex-5p::mito(tomm-20$^{1-54}$)-Dendra2::nos-2 3'UTR)] I; xnSi45 [pYA11(mex-5p::mCherry-moma-1::nos-2 3'UTR, unc-119(+))] II; atg-18(gk378) V; uaDf5 /+ mtDNA* | This study | FT2417 | Shown in **Figure 5F**. |
| Sequence-based reagent | *ocrAS_Dendra-C-term* | Integrated DNA Technologies (IDT) | GTCCTCTACCAAGTCAAGCA | crRNA to replace Dendra in *xnSi67* |
| Sequence-based reagent | *ocrAS_Dendra-N-Term* | IDT | AGAATGTCGGACACAATTCT | crRNA to replace Dendra in *xnSi67* |
| Sequence-based reagent | ocrAS01 | IDT | AAGGGAGAAGAATTATTTAC | crRNA used to add MLS to GFP$_{1-10}$ in *xnSi73* |
| Sequence-based reagent | ocrAS13 | IDT | ATCTGCATTTTCTTTCTGTT | crRNA used for *hmg-5 C-terminal* tagging |
| Sequence-based reagent | ocrAS19 | IDT | GGTGATAAATGGGTTTGAGA | crRNA used for *dct-1(STOP-IN)* |
| Sequence-based reagent | ocrAS20 | IDT | CAGGTGTACTCTCGGTCAAT | crRNA used for *dct-1(STOP-IN)* |
| Sequence-based reagent | ocrAS25 | IDT | AACTCCTAAATTATAAGTGG | crRNA used for *pink-1(STOP-IN)* |
| Sequence-based reagent | ocrAS26 | IDT | ATGAACTCCTAAATTATAAG | crRNA used for *pink-1(STOP-IN)* |
| Sequence-based reagent | oAS115 | IDT | TTTATCGATAATCAATTGA ATGTTTCAGACAGAGAAT GGCACTCCTGCAATCAC GTCTCCTCCTGTCCGCC CCACGTCGTGCCGCCG CCACCGCCCGTGCCGG AGCTGGTGCAGGCGCT GGAGCCGGAGCCATGT CTAAGGGAGAAGAACT CTTCACTGGAGTTGTT CCTATCCTCGTCGAGC TCGACGGAGACG | MLS-GFP$_{1-10}$ repair template |
| Sequence-based reagent | oAS187 | IDT | tttgattacaaaatggaaag ttgtgacgaattcaaCTAG GTGATTCCGGCGG CATTGACATACTCA TGGAGGACCATGT GGTCACGTCCTCC TGAACCTCCTTGAT CTGCATTTTCTTTT TGTTCTGCTTCCC ATTTCTGGAGGAC GACATGGTATTCATCT | *hmg-5-GFP$_{11}$* repair template |
| Sequence-based reagent | oAS216 | IDT | aaaaagtaaaacaaac CAGGTGTACTCT CGGTCAAGCTAG CTTATCACTTAGT CAAGCATAATCTG GAACATCATATGG ATAAGCGTAGTCT GGAACGTCGTATG GATATGCATAGTCT GGCACGTCGTATG GGTAGACGGCTTT TGCGGATGGTGTT GTCTGTTGAGCCG | *dct-1(STOP-IN)* repair template |

*Continued on next page*

*Continued*

| Reagent type (species) or resource | Designation | Source or reference | Identifiers | Additional information |
|---|---|---|---|---|
| Sequence-based reagent | oAS245 | IDT | GAGCCTTTTTGAG TACGACATGAACT CCTAAATTAGCTA GCTTATCACTTAG TCACCTCTGCTCT GGACAAACTTCCC TCCTCCTGAACCT CCCGATGCTCCTG AGGCTCCCGATGC TCCTAAGTGGCGG GAAATATTCTCGGC AGGAAGCGTTG | *pink-1(STOP-IN)* repair template |

## Worm culture and strains

Unless otherwise stated, all strains were maintained at 20°C on nematode growth medium plates seeded with *Escherichia coli* strain OP50 according to standard methods (*Brenner, 1974*). For egg isolation and L1 synchronization, semi-synchronized L1 larvae were outgrown on 10 cm enriched peptone plates seeded with *E. coli* strain NA22. Gravid adults were then washed off and early-stage embryos were isolated via worm bleaching. Isolated eggs were broken into two populations: one for immediate embryo dissociation and another which was allowed to hatch and starved overnight in M9 for L1 synchronization/dissociation. For late embryo dissociations, early embryos were isolated as above and incubated in M9 at 25°C for 6 hr. For L1 feeding experiments, synchronized L1 larvae were plated onto enriched peptone plates and grown for 12 and 24 hr at 20°C (for cell sorting), or for 6, 9, 12, and 24 hr at 23°C (for live imaging). A list of all strains used/generated in the study is available in the Key resources table.

## PGC isolation and cell sorting

Cell dissociation of embryos and larvae was performed as described previously (*Lee et al., 2017*; *Strange et al., 2007*) with slight modifications described in detail below.

### Embryonic cell dissociation

Purified embryos were pelleted at 3000 × *g* for 30 s in non-stick 1.5 mL tubes (Thomas Scientific 1149X75), resuspended in 600 µL chitinase (Sigma C6317; 2 mg/mL) in conditioned-egg buffer (25 mM HEPES [Sigma H3375] pH 7.3, 118 mM NaCl, 48 mM KCl, 2 mM CaCl$_2$, 2 mM MgCl$_2$, adjusted to mOsm 340±5 with ddH$_2$O), hereafter referred to as egg buffer, and incubated on a rocking nutator for 15 min at room temperature. After 15 min, 800 µL of cold egg buffer was added, embryos were spun at 900 × *g* for 4 min at 4°C, and then resuspended in 800 µL Accumax-egg buffer solution (Innovative Cell Technologies, AM105, 1:3 dilution ratio in egg buffer). For dissociation, embryos were pipetted up and down ~80 times using a P1000 pipette. To wash away debris, dissociated embryos were spun at 900 × *g* for 4 min at 4°C a total of three times. Washed cells were resuspended in 800 µL of cold egg buffer, and single cells were separated from clumps by gravity settling on ice for 15–20 min. For *uaDf5* heteroplasmy experiments, 25 µL of dissociated cells were removed at this stage, mixed 1:1 with worm lysis buffer, lysed as described below, and stored at –80°C for ddPCR.

### Late embryonic cell dissociation

To isolate late embryos (majority above 1.5-fold/2-fold), purified early-stage embryos were isolated as above, and incubated in M9 at 25°C with rotation for 6 hr. After aging, late embryos were then collected into a 15 mL conical tube and spun at 3000 × *g* for 30 s. Pelleted eggs were transferred into non-stick 1.5 mL tubes, spun at 3000 × *g*, and washed 1× with 1 mL M9 then 2× with 1 mL egg buffer. Eggs were resuspended in 600 µL chitinase (see above) and incubated for 10 min at room temperature. After 10 min, 800 µL of cold egg buffer was added and late embryos were spun at 3000 × *g* for 30 s and washed an additional 2× with egg buffer. Eggs were resuspended in 250 µL SDS-DTT solution (20 mM HEPES pH 8.0, 0.25% SDS (sodium dodecyl sulfate) [Sigma 71725], 200 mM DTT (dithiothreitol) [Sigma D0632], 3% sucrose), and incubated for 1 min at room temperature with gentle mixing. To stop the reaction, 1 mL of cold egg buffer was added, then animals were spun at 16,000 × *g* for

1 min and washed an additional 5× with cold egg buffer. Following the last wash, SDS-DTT treated embryos were resuspended in 250 µL pronase (Sigma P8811) solution (15 mg/mL in egg buffer) and dissociated by pipetting up and down 80–120 times, using a P200 pipette, over the course of 5 min. To end the dissociation, 1 mL of cold egg buffer was added, and cells were spun down at 1600 × g for 6 min at 4°C. Cell pellets were resuspended in 1 mL of cold egg buffer and washed an additional 3× by spinning 1600 × g for 6 min at 4°C. Following the final wash, dissociated cells were resuspended in 800 µL of cold egg buffer and separated from undissociated embryos and clumps by gravity settling on ice for 30–40 min.

## Larval cell dissociation

Dissociation of larvae was performed at three stages: starved L1s, mid-L1s (L1s fed 12 hr, 20°C), and L2s (L1s fed 24 hr, 20°C). Larvae at a specific stage were collected into 15 mL conical tubes, spun down at 3000 × g for 30 s, and washed with ddH$_2$O 2–6×. Larvae were then collected in 1.5 mL non-stick tubes and spun at 16,000 × g for 2 min. Depending on the size of the pellet, larvae were split into multiple tubes such that each tube had no more than 100 µL of pelleted animals. Starved L1s, mid-L1s, and L2s were then resuspended in 250 µL of SDS-DTT solution (see above) and incubated for 2, 2.5, and 3 min, respectively with gentle mixing. To stop the reaction 1 mL of cold egg buffer was added, then animals were spun at 16,000 × g for 1 min and washed an additional 5× with cold egg buffer. Following the last wash, SDS-DTT treated animals were resuspended in 250 µL pronase solution (see above) and incubated for 5–15 min on a rocking nutator at room temperature. Animals were then dissociated by trituration with a P200 pipet for an additional 10–25 min (~60 times every 5 min) in pronase solution. To end the dissociation, 1 mL of cold egg buffer was added and cells were spun down at 9600 × g for 3 min at 4°C. Cell pellets were resuspended in 1 mL of cold egg buffer and washed 3× by spinning 1600 × g for 6 min at 4°C. Following the final wash, dissociated cells were resuspended in 800 µL of cold egg buffer and separated from undissociated larvae and clumps by gravity settling on ice for 30–40 min.

## FACS and PGC isolation

For sorting experiments, we used a strain expressing endogenously tagged GLH-1-GFP, which is a germline-specific protein (*Marnik et al., 2019*), as well as a transgenic mCherry marker (*xnIs510*) specific to somatic gonad precursor cells (SGPs) (*McIntyre and Nance, 2020*), which ensheath the PGCs and are the most likely contaminating population of cells. Approximately 15 min prior to cell sorting, DAPI (4',6-diamidino-2-phenylindole,Sigma D9542) was added to the cells (final concentration of 0.125 µg/mL) as a viability marker. The GLH-1-GFP(+); SGP-mCherry(-); DAPI(-) cells were isolated via FACS using a 100 µm nozzle on a BD FACSAria II cell sorter. Singlet cells were sorted for all samples except for *nop-1* L1 PGCs, which are born binucleate and cellularize following the first PGC cell division. For quality control, sorted cells were live imaged (see 'Microscopy' below) to confirm the presence of GFP(+); mCherry(-) cells. Purity was assayed, via post-sort analysis, by resorting cells and quantifying the percentage of GFP(+); mCherry(-); DAPI(-) cells in the population using FlowJo software V10 (embryo: 98.0% ± 0.5 pure [N=3]; L1: 97.5% ± 2.7 pure [N=3]). For most ddPCR analyses, 1000–5000 PGCs were sorted into 500 µL of 0.5× worm lysis buffer (recipe below) in a screw-cap 1.5 mL microfuge tube (20,000 and 10,000 cells were sorted for wild-type and *TFAM-GFP* PGCs, respectively). Following sorting, PGCs were lysed for 30 min on ice and then incubated in a tabletop heating block for 1 hr at 55°C followed by 15 min at 95°C. Cell lysates were frozen at –80°C until needed for ddPCR. For live imaging, 1000–2500 PGCs were sorted into 500 µL of conditioned L-15 medium (10% FBS (fetal bovine serum), 50 U/mL penicillin + 50 µg/mL streptomycin [Sigma P4458], adjusted to mOsm 340 ± 5 with 60% sucrose) and kept on ice. Embryonic and larval PGCs were spun down at 900 × g (4 min) and 1600 × g (6 min), respectively, all but 50 µL of conditioned L-15 was removed, and cells were gently resuspended for imaging (see 'Microscopy' below).

## qPCR of L4 larvae

For standard curve generation, an 887 bp portion of mtDNA containing *nd-1* was amplified by PCR and cloned into pMiniT2.0 using the NEB PCR cloning kit (NEB E1202S). The purified plasmid was linearized with BamHI-HF (NEB 3136), and DNA concentration was quantified using a Nanodrop spectrometer (Thermo Scientific). For the standard curve, 64,000, 32,000, 24,000, 16,000, 12,000,

8000, 6000, and 4000 copies of plasmid were run in triplicate as described below. Oligos targeting the mitochondrial gene *nd-1* (see 'ddPCR' below) were used for qPCR quantification. For absolute quantification, single late-L4 larvae were picked into 5 µL of worm lysis buffer (50 mM KCl, 10 mM Tris-HCl [pH 8.0], 2.5 mM MgCl$_2$, 0.45% IGEPAL [Sigma I8896], 0.45% Tween 20 [Sigma P9416], 0.01% gelatin [Sigma G1393], and 200 µg/mL proteinase K [Invitrogen 2530049] and flash frozen at –80°C for 15 min). Worms were then lysed in a thermal cycler at 60°C for 1 hr followed by 15 min at 95°C. Prior to qPCR, lysed L4s were diluted 20× by adding 95 µL of nuclease-free water (Invitrogen 4387936) and mixed thoroughly by pipetting. About 8 µL of the lysate (or diluted plasmid for standard curve) was used in triplicate for each individual sample. The qPCR was performed as a 20 µL reaction with 500 µM of each primer, using BioRad 2× SsoAdvanced Universal SYBR Green Supermix (BioRad 1725271) in a Roche LightCycler 480 machine. The PCR program was as follows: 10 min at 98°C, 40 cycles of 98°C for 15 s, and 60°C for 1 min. Crossing point values were derived using the Second Derivative Maximum method of the Roche LightCycler 480 software.

## Whole embryo lysis

Embryos were isolated from gravid adults and treated with chitinase for 8 min at room temperature to dissolve the eggshell prior to lysis. Chitinase-treated embryos were washed 2–3× with cold egg buffer and transferred to a watch glass. Exactly four early-stage embryos (pre-bean stage) were mouth-pipetted into 20 µL worm lysis buffer per tube using a hand-pulled glass capillary. Embryos were then lysed in a thermal cycler (as above) and stored at –80°C.

## Droplet digital PCR (ddPCR)

Prior to ddPCR, various sample types were diluted to different degrees in nuclease-free water: sorted-PGC lysates (4×), dissociated-embryo lysates (3000–6000×), whole-embryo lysates (10×), and whole-adult lysates (30 adults lysed in 60 µL lysis buffer, 1000×). The ddPCR was run according to the manufacturer's recommendations. Briefly, ddPCR reactions were assembled as 24 µL mixes containing 0.1 µM of each primer, Bio-Rad QX200 ddPCR EvaGreen Supermix (BioRad 186–4034), 0.1 U/µL SacI-HF (New England Biolabs), and 4.8 µL of the sample. Reactions were incubated in the dark at room temperature for 30–60 min to allow SacI-HF (NEB R3156) digestion to linearize/digest DNA prior to droplet generation. After incubation, samples were loaded for droplet generation in a BioRad QX200 Automated Droplet Generator. The PCR amplification was performed as follows: 10 min at 95°C, 40 cycles of 94°C for 30 s, and 60°C for 1 min, followed by 10 min at 98°C for all primer pairs. Samples were all run in triplicate and were immediately analyzed using a BioRad QX200 Droplet reader. All ddPCR reactions were single oligo-pair mixes; therefore, absolute DNA concentrations were calculated using 1D-amplitude plots in BioRad QuantaSoft software.

## mtDNA copy number quantification

The absolute mtDNA copy number per cell was determined using primer pairs targeting mtDNA (*nd-1*) and gDNA (*cox-4*).

mtDNA –

nd-1_Fw: 5'- agcgtcatttattgggaagaagac –3'
nd-1_Rv: 5'- aagcttgtgctaatcccataaatgt –3'

cox-4_Fw: 5'- gccgactggaagaacttgtc –3'
cox-4_Rv: 5'- gcggagatcaccttccagta –3'

Two independent ddPCR reactions of the same sample were run simultaneously to determine the mtDNA copies/µL and gDNA copies/µL. The mtDNA copy number/cell was calculated as follows:

total mtDNAs detected **/** [total gDNA detected **/** (N)],

where the ploidy (N)=4 since *C. elegans* PGCs are arrested in the G2 phase of the cell cycle (*Fukuyama et al., 2006*). For L1 feeding experiments, the ploidy was calculated based on the expected versus the actual number of gDNAs detected (*Figure 3—figure supplement 2E*). Since the ploidy of starved L1 PGCs is constant, we could normalize our data as such. For example, we found that when we sorted 5000 starved L1 PGCs we detected 61 gDNA copies via our ddPCR assay. Therefore, when we sorted 5000 mid-L1 or L2 PGCs and only detected 46 gDNAs we estimated the ploidy as follows:

[(actual copies detected: 46) **/** (expected copies detected: 61)] × 4,

where the multiplication factor 4 adjusts the ratio with respect to N=4 for starved L1 PGCs. Thus, for fed L1/L2 PGCs the ploidy (N) can be estimated as approximately 3. This value agrees well with estimated ploidy values based on the calculated cell cycle occupancy times of mitotic germ cells in *C. elegans* adults (*Fox et al., 2011*). To calculate mtDNAs/germline, the value for mtDNAs/cell was multiplied by the average number of observed germ cell nuclei at the corresponding stage (see 'PGC/GSC counts' below).

### ΔmtDNA (*uaDf5* and *mptDf2*) heteroplasmy quantification

mtDNA heteroplasmy was determined using four oligo pairs that specifically detect *uaDf5, mptDf2,* and their respective complementing WT mtDNAs:

For *uaDf5* heteroplasmy –

uaDf5-mtDNA_Fw: 5'- ccatccgtgctagaagacaaag –3'
uaDf5-mtDNA_Rv: 5'- ctacagtgcattgacctagtcatc –3'
WT-mtDNA_Fw: 5'- gtccttgtggaatggttgaatttac -3'
WT-mtDNA_Rv: 5'- gtacttaatcacgctacagcagc -3'

For *mptDf2* heteroplasmy –

mptDf2-mtDNA_Fw: 5'- ggattggcagtttgattagagag –3'
mptDf2-mtDNA_Rv: 5'- aagtaacaaacactaaaactcccaac –3'
WT-mtDNA_Fw: 5'- cgtgcttatttttcggctgc -3'
WT-mtDNA_Rv: 5'- ctttaacacctgttggcactg -3'

Two independent ddPCR reactions were run simultaneously for each sample to determine the WT mtDNA copies/μL and mutant mtDNA copies/μL. Percent heteroplasmy was then calculated as follows:

[ΔmtDNA **/** (ΔmtDNA +WT mtDNA)] × 100.

## Microscopy

Embryos, adults, and larvae were mounted on 5 and 10% agarose pads, respectively. Larvae were immobilized prior to and during image acquisition using 1.25 mM levamisole in M9 buffer. Animals were imaged on a Leica SP8 laser-scanning confocal microscope, using a 63 × 1.4 NA oil-immersion objective with 488 nm and 594 nm lasers and HyD detectors; or on a Zeiss AxioImager A2, using a 40 × 1.3 NA oil-immersion objective and a charge-coupled device (CCD) camera (model C10600-10B-H, S. 160522; Hamamatsu). For sorted PGC imaging, 5 μL of sorted embryonic and larval PGCs in conditioned L-15 (see 'FACS and PGC isolation' above) were mounted on custom depression slides to avoid crushing the cells. Sorted PGCs were then imaged on a Zeiss AxioImager A2 as above. Images were analyzed and processed in ImageJ (NIH), and Adobe Photoshop.

## Image analysis

### Mitochondrial acidification

Acidification of mitochondria was measured in embryos and L1 larvae by determining the ratio of green-to-red fluorescence of Mito-mCh$^{PGC}$ and Mito-Dendra$^{PGC}$. For L1 larvae, 488 nm and 594 nm laser intensities were adjusted to ensure a similar dynamic range of signal intensity for Mito-mCh$^{PGC}$ and Mito-Dendra$^{PGC}$ within the PGC cell body. Two regions of interests (ROIs) were drawn – one around PGC lobe debris and the other around cell body mitochondria. Red and green signal intensity was then measured and analyzed using ImageJ (NIH) software.

Acidified mitochondria in the embryo were defined as regions of the PGC mitochondrial network where the red signal overtook green, such that the measured green-to-red signal ratio was at least twofold less compared to the greater mitochondrial network (*Figure 2—figure supplement 4*). The PGCs of 1.5-fold to 2-fold embryos were imaged and scored categorically as either containing or not containing, regions of acidified mitochondria. An ROI was then drawn around regions with a red dominant signal, and green/red signal intensity was measured in ImageJ. A green/red signal was also measured within an ROI enclosing the rest of the mitochondrial network for comparison.

## TFAM-GFP colocalization with Mito-mCh$^{PGC}$

Adult *C. elegans* were mounted on 5% agarose pads and imaged by confocal microscopy as above. The fraction of TFAM-GFP that colocalized with mitochondria (Mito-mCh$^{PGC}$) was calculated in a single Z-plane by drawing a region of interest around the distal adult germ line and measuring Manders' Colocalization Coefficient using the plugin 'JACoP' in ImageJ (NIH).

## Quantification of mitochondrial localization in PGCs

One-and-a-half-fold and two-fold embryos were imaged as described above. Mitochondrial content was measured as a sum of Mito-Dendra$^{PGC}$ positive voxels within the PGC using ImageJ. An ROI was then drawn specifically around the PGC cell body using Mem-mCh$^{PGC}$ as a marker, and the fraction of mitochondria in the PGC cell body was calculated as a ratio of total PGC mitochondria.

## In vivo measurement of embryonic PGC and whole embryo volume

The volume of PGCs was determined in embryos just prior to lobe formation (bean stage) and in starved L1 larvae. A Z-stack was taken through the PGCs of animals expressing a PGC-specific plasma membrane marker (*xnSi1*; *Chihara and Nance, 2012*), and the volume of both PGCs was measured by defining the PGC surfaces using the image analysis platform Imaris (Oxford Instruments); the volume contained within them was measured and divided by two to determine the volume per single PGC. Embryo volume was calculated by measuring the anterior-posterior and left-right axes of fertilized embryos in ImageJ. Whole embryos were assumed to approximate an ellipsoid, and the volume was calculated using the formula $V = 4/3\ \pi\ a \times b \times c$, where a, b, and c are the radii of the three axes of the ellipsoid (the width and height of embryos were assumed to be equal).

## Quantification of TFAM foci

Embryos, starved L1, early-L1, mid-L1, late-L1, and L2 larvae were mounted as described above (see 'Microscopy'). A full Z-stack of the entire germline was taken for each animal. Germline TFAM-GFP/ GFP$_{11}$ foci were identified using ImageJ to segment TFAM-GFP/GFP$_{11}$ signal that colocalized with Mito-mCh$^{PGC}$. Colocalized TFAM-GFP/GFP$_{11}$ foci were then defined as local signal maxima and relative numbers of foci were counted using the 3D maxima plugin of the ImageJ 3D suite.

## PGC/GSC counts

Embryos and starved L1 larvae were assumed to have exactly two PGCs. For fed larvae expressing TFAM-GFP/GFP$_{11}$ and Mito-mCh$^{PGC}$, the number of cells per animal was determined by counting the dark spots in image stacks surrounded by Mito-mCh$^{PGC}$ as a proxy for germ cell nuclei. For cell sorting experiments, fed larvae were mounted and imaged just prior to cell dissociation (see 'Larval cell dissociation' above), and germ cell counts were determined by counting the number of nuclei surrounded by GLH-1-GFP.

## Ex vivo measurement of sorted PGC volume

Sorted embryonic and L1 PGCs were imaged as described above (see 'Microscopy'). The PGC diameter was calculated by drawing a line across the center of the cell and measuring its length in ImageJ. The PGC volume was determined under the assumption that the PGCs approximate a sphere, and volume was calculated with the formula $V = 4/3\pi r^3$.

## Transgene construction

Transgenes *mex-5p::tomm-20$^{1-54}$-Dendra2::nos-2 3'UTR, unc-119(+)* (plasmid *pYA57*) and *mex-5p::GFP$_{1-10}$::nos-2 3'UTR, unc-119(+)* (plasmid *pAS07*) were constructed by Gibson assembly (*Gibson et al., 2009*). Briefly, overlapping primers were used to amplify *tomm-20$^{1-54}$-Dendra2* to replace *mCherry-moma-1* in *pYA11 (mex-5p::mCherry-moma-1::nos-2 3'UTR, unc-119(+))*, a derivative of *pCFJ150*. Split *GFP$_{1-10}$* was *C. elegans* codon-optimized, designed with introns and ordered as a gBlock (IDT) with overhangs to replace *mCherry-PH* in *pAS06 (mex-5p:: mCherry-PH::nos-2 3'UTR, unc-119(+))*, a derivative of *pCFJ150* that lacks a portion the universal MosSCI homology sequence to facilitate CRISPR mediated insertion of the plasmid (*Dickinson et al., 2013*).

## Transgenesis and genome editing

### MosSCI

*pYA57 (mex-5p::tomm-20$^{1-54}$-Dendra2::nos-2 3'UTR, unc-119(+))* was microinjected into strain EG8078 to create *xnSi67*, a single-copy insertion on chromosome I, via the Universal MosSCI method (***Frøkjaer-Jensen et al., 2008***).

### CRISPR/Cas9

In all cases, CRISPR/Cas9 mediated genome editing was performed using pre-incubated Cas9 (Berkeley)::(crRNA +tracrRNA) (IDT) ribonucleoprotein, and injection quality was screened using the co-CRISPR *dpy-10(cn64)* mutation as previously described (***Paix et al., 2017***). DNA repair templates contained ~25–35 bps of homology on each arm, and varied depending on the size of insertion as either dsDNA PCR product (>150 bps), or ssDNA oligos (<150 bps) (IDT). The crRNAs and insertion sequences are listed in the Key resources table and ***Supplementary file 1***. For the generation of putative null alleles [*pdr-1(xn199), dct-1(xn192)*] we used the 'STOP-IN' method (***Wang et al., 2018***) to insert an early stop and frame-shift into either the first or second, exon of the target gene. For the generation of *xnSi73 [mex-5p::GFP$_{1-10}$::nos-2 3'UTR, unc-119(+)]*, *pAS07* was used as a PCR template to amplify *GFP$_{1-10}$* with ~35 bp of homology to replace *tomm-20$^{1-54}$::Dendra2* by CRISPR at the *xnSi67* locus. To generate *xnSi85 [mex-5p::mito(matrix)GFP$_{1-10}$::nos-2 3'UTR, unc-119(+)]*, an oligo repair template (see Key resource table) was used to introduce an N-terminal mitochondrial-matrix localization sequence to *xnSi73*. To generate *hmg-5(xn107[hmg-5-GFP])* and *hmg-5(xn168[hmg-5-GFP$_{11}$])*, full length *GFP* with ~35 bp homology arms or an oligo-containing sequence for *GFP$_{11}$* were used to generate C-terminal tags at the endogenous *hmg-5* locus. All strains generated by CRISPR are included in the Key resources table and relevant sequences are in ***Supplementary file 1***.

## Statistical analysis and reproducibility

Statistical analysis was performed using GraphPad Prism 9 software. For categorical data, such as scoring acidified mitochondria in PGCs, contingency tables were made and Fisher's exact test was used to calculate p-values. For all other data, one-tailed or two-tailed Student's *t*-tests were performed, as applicable. For mtDNA copy number comparisons, unpaired *t*-tests were used since embryos and L1s could come from the same or different adult populations; for heteroplasmy experiments, paired tests were used since embryos and L1 PGCs always came from the same adult population. Data in graphs are shown as Superplots (***Lord et al., 2020***), with individual data points from three independent color-coded biological replicates (except for ddPCR experiments where small dots are technical replicates of the ddPCR analysis) shown as small dots, the mean from each experiment shown as a larger circle, the mean of means as a horizontal line, and the SEM as error bars. Sample size, *t*-test type, and p-value ranges are reported in figure legends. Where applicable, no corrections for multiple comparisons were made to avoid type II errors (***Armstrong, 2014***). For live imaging, embryos and larvae were selected based on orientation on the slide and on health. For all datasets, at least three biologically independent experiments were performed and the arithmetic means of biological replicates were used for statistical analysis. Combined source data for all ddPCR experiments can be found in ***Supplementary file 2***.

## Acknowledgements

We thank the *Caenorhabditis* Genetics Center (CGC), Heng-Chi Lee (U. of Chicago) and Dustin Updike (MDI Biological Laboratory) for providing worm strains. The CGC is supported by the NIH Office of Research Infrastructure Programs (P40 OD010440). We thank members of the Nance laboratory, Ruth Lehmann, Florenal Joseph, and Melissa Pamula for comments on the manuscript. We thank Peter Lopez, James Alvarado, Yulia Chupalova, and Sitharam Ramaswami for FACS/ddPCR assay development, Michael Cammer and Yan Deng for help with image analysis and acquisition, and Ibrahim Abdel Wahab for analyzing PGC volumetric data. FACS was performed at the NYULMC Cytometry and Cell Sorting Laboratory; ddPCR was performed at the NYULMC Genome Technology Center; and microscopy used instrumentation in the NYULMC Microscopy Laboratory, all of which are partially supported by the Laura and Isaac Perlmutter Cancer Center support grant P30CA016087 from the National Institutes of Health/National Cancer Institute. This work was supported by a training grant from NYSTEM

(C32560GG) and a fellowship from the National Institutes of Health (NIH) (F31HD102161) to AZAS; a research grant from the NIH (R01GM123260) and a Discovery Grant from CDMRP (PR170792) to MRP; and a research grant from the NIH to JN (R35GM118081).

## Additional information

### Funding

| Funder | Grant reference number | Author |
|---|---|---|
| New York State Stem Cell Science | C32560GG | Aaron ZA Schwartz |
| Eunice Kennedy Shriver National Institute of Child Health and Human Development | F31HD102161 | Aaron ZA Schwartz |
| National Institute of General Medical Sciences | R35GM118081 | Jeremy Nance |
| National Institute of General Medical Sciences | R01GM123260 | Maulik R Patel |
| Congressionally Directed Medical Research Programs | PR170792 | Maulik R Patel |

The funders had no role in study design, data collection and interpretation, or the decision to submit the work for publication.

### Author contributions

Aaron ZA Schwartz, Conceptualization, Resources, Data curation, Formal analysis, Investigation, Methodology, Writing – original draft, Writing – review and editing; Nikita Tsyba, Methodology, Writing – review and editing; Yusuff Abdu, Maulik R Patel, Resources, Writing – review and editing; Jeremy Nance, Conceptualization, Resources, Formal analysis, Methodology, Writing – original draft, Writing – review and editing

### Author ORCIDs

Aaron ZA Schwartz  http://orcid.org/0000-0001-9559-8903
Maulik R Patel  http://orcid.org/0000-0003-3749-0122
Jeremy Nance  http://orcid.org/0000-0003-4212-7731

### Decision letter and Author response

Decision letter https://doi.org/10.7554/eLife.80396.sa1
Author response https://doi.org/10.7554/eLife.80396.sa2

## Additional files

### Supplementary files
• Supplementary file 1. Supplemental sequences.
• Supplementary file 2. Combined ddPCR source data.
• MDAR checklist

### Data availability

All data generated or analyzed during this study are included in the manuscript and supporting files. Source data files have been provided for Figures 1-6 and accompanying Figure Supplements.

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
