## [Editor Report]

Mitochondria have their own DNA, which is much more likely to gain mutations (due to error-prone DNA polymerase). It is widely appreciated that there are quality control mechanisms such that functional mitochondria are passed from one generation to the next. This manuscript presents important progress in the field, describing how the *C. elegans* germline may remove mitochondria by creating bottlenecks as well as selectively removing non-functional mitochondria. Building upon the authors' previous finding that the *C. elegans* primordial germ cells (PGCs) shed much of their cytoplasm during embryogenesis through 'cannibalism', they now describe that a bulk of mitochondria are removed from PGCs through this process. Although some of the phenotypes described in the manuscript are relatively mild, the evidence is compelling, supporting their conclusions.

---

## [Decision Letter]

**Decision letter after peer review:**

Thank you for submitting your article "Independent regulation of mtDNA quantity and quality resets the mitochondrial genome in *C. elegans* primordial germ cells" for consideration by *eLife*. Your article has been reviewed by 3 peer reviewers, including Yukiko M Yamashita as Reviewing Editor and Reviewer #1, and the evaluation has been overseen by Benoît Kornmann as the Senior Editor.

Essential revisions:

1) Assessing mtDNA replication more directly would strengthen the manuscript. This could be done by assessing EdU/BrdU incorporation in the presence of nuclear replication inhibitors. This could be technically difficult, however: if it is the case, textual changes to slightly tone down the statement in the manuscript would suffice.

2) It would be informative to show the L1/2 stage data for the nop1 mutant in Figure 3j and l. The late GSCs of wt and nop1 mutant carry a similar number of mtDNA copies while their starting numbers are very different. Given that the major part of the manuscript is to quantify the mtDNA copy number during the embryonic PGCs, and PGCs from the embryonic to larval transition, understanding how compromising cannibalism affects the mtDNA copy number during different stages will make the paper stronger. Including an analysis of mutant and wildtype mtDNA copy number could be informative, as this might provide more information about how selection works during the embryo-larval transition in PGCs.

Please note that individual reviewer comments are provided in their entirety to assist your revision. However, you are not required to fully address the matters that are not listed as essential revisions here. They are provided for your reference.

*Reviewer #1 (Recommendations for the authors):*

– line 177: 'gfp(11)' it took a while for me to realize this is a part of GFP used for BiFC – just because I couldn't tell what '(11)' was referring to (I even thought (11) was referring to reference…). Can you add a bit of explanation so that there will be no confusion? E.g. 'tagged with a fragment of GFP ('GFP(11)') that is used for BiFC assay'.

– the term 'purifying selection' feels a bit too strong for only ~50% reduction in the number and only ~5% reduction in heteroplasmy. It is fine to correlate the current work with purifying selection, but probably better not to call the observation 'purifying selection'.

*Reviewer #3 (Recommendations for the authors):*

I have the following recommendations:

1. With respect to mtDNA replication in PGCs, this could be addressed experimentally by measuring mtDNA replication using EdU, although this may be technically challenging and not possible in their system. They could address this textually by being clearer throughout the text that they are not measuring replication, and perhaps discussing how, if replication is occurring in PGCs, this would influence their interpretation of their data.

2. In the introduction, consider describing in greater detail worm germline development and cannibalism.

3. The Mito-GFP(1-10)PGC + TFAM-GFP(11) looks diffuse and not punctate (see figure S4). Given this, how did the authors count GFP puncta in this strain? Consider including a more detailed explanation of this quantification in the methods.

4. On lines 296 -297 the authors state "It is possible that this number of mtDNAs is needed for sufficient selection against deleterious mtDNA mutations". Does the fact that selection is not impaired in *nop-1* mutants suggest otherwise? If so, consider removing this statement.

---

## [Author Response]

Essential revisions:1) Assessing mtDNA replication more directly would strengthen the manuscript. This could be done by assessing EdU/BrdU incorporation in the presence of nuclear replication inhibitors. This could be technically difficult, however: if it is the case, textual changes to slightly tone down the statement in the manuscript would suffice.

This is an excellent suggestion – we agree that directly observing mtDNA replication in PGCs would be a powerful complement to our ddPCR and imaging experiments. As proposed, we attempted to do this by EdU labeling newly hatched L1 larvae. As a positive control to see if we could detect replicated mtDNAs in germ cells, we incubated L1 larvae with EdU and fed for 6 hours, which our experiments imaging TFAM-GFP_11_ showed is a sufficient time for mtDNA number to significantly increase in the germ line due to replication. While a few replicating nuclei within fed larva were labeled with EdU, we saw no convincing EdU signal outside of nuclei (see Author response image 1). Because of these disappointing results, and the significant time investment potentially involved in increasing the sensitivity of this approach to detect mtDNA replication in *C. elegans* for the first time, we addressed PGC mtDNA replication in other quantitative ways.

**Author response image 1. sa2fig1:** 

We performed two additional experiments on FACs-sorted PGCs to quantify mtDNAs and look for evidence of mtDNA replication. First, we sorted and analyzed PGCs from late embryos, which were six hours older than the pre-cannibalism population of embryonic PGCs we analyzed previously. Since these PGCs have just recently completed lobe cannibalism, but are significantly younger than those we purified from L1 larvae, ongoing mtDNA replication should result in lower levels of mtDNA in late embryo PGCs compared to L1 PGCs. Interestingly, we found that while late embryo PGC mtDNAs were reduced relative to our previously analyzed pre-cannibalism embryo PGCs as expected, mtDNA levels per PGC were slightly but significantly *higher* than in L1 PGCs, even though lobe cannibalism was fully completed at this stage as assessed by analysis of the volume of sorted cells (data added to Figure 2B as “late emb”; FACS gating strategy and cell volume data included in Figure 2—figure supplements 1 and 2). Because we showed previously that autophagy operates in PGCs and is responsible for eliminating some mtDNAs, we suspect that the lower number of mtDNAs in L1 PGCs versus late embryonic PGCs is the result of ongoing bulk autophagy (see Figure 5). This new experiment suggests either that mtDNA replication does not occur in embryonic PGCs, or that it is minimal in its extent and more than offset by concurrent autophagy.To directly test this, in a second experiment, we sorted embryonic and L1 PGCs in *nop-1; atg-18* double mutants. Because these mutants lack both lobe cannibalism and autophagy, the number of mtDNAs in L1 PGCs should be the same as in embryonic PGCs unless it is increased by mtDNA replication. We found that the number of mtDNAs in *nop-1; atg-18* L1 PGCs was not statistically higher than that in *nop-1; atg-18* embryonic PGCs, but these mutants inherited a significantly higher proportion of embryonic PGC mtDNAs than *nop-1* single mutants (data added to Figure 2E,G), suggesting that autophagy and lobe cannibalism are both required for the complete reduction of mtDNA in PGCs. Together with our existing data, these two additional experiments support the conclusion that robust mtDNA replication does not occur in PGCs until L1 larvae begin to feed; if mtDNA replication does occur in PGCs prior to this stage, it is very minimal and below our ability to detect by ddPCR.

Because neither of these experiments completely rules out the possibility of selective replication of a small subset of mtDNAs in embryonic PGCs, we altered the language in the text to leave this possibility open. Specifically, rather than refer to the initiation of *“mtDNA replication”* occurring during the PGC-to-GSC transition, we now refer to this as the initiation of *“mtDNA expansion”* as this is the first point in development that we see total germline mtDNA numbers increase after lobe cannibalism and autophagy reduces them (see for example pg. 10, lines 251-254). Also, based on these new findings, we felt that introducing autophagy earlier in the paper – in this section rather than waiting until the section on purifying selection – would be more logical. These textual rearrangements are reflected in the version of the manuscript with changes tracked.

2) It would be informative to show the L1/2 stage data for the nop1 mutant in Figure 3j and l. The late GSCs of wt and nop1 mutant carry a similar number of mtDNA copies while their starting numbers are very different. Given that the major part of the manuscript is to quantify the mtDNA copy number during the embryonic PGCs, and PGCs from the embryonic to larval transition, understanding how compromising cannibalism affects the mtDNA copy number during different stages will make the paper stronger. Including an analysis of mutant and wildtype mtDNA copy number could be informative, as this might provide more information about how selection works during the embryo-larval transition in PGCs.

Thank you for suggesting this experiment. We sorted *nop-1* mutant germ cells at the mid-L1 stage to see if germline mtDNA numbers had already begun to normalize back to ~200 per GSC. Mid-L1 *nop-1* mutants contained on average four GSCs. We found that even at this early timepoint, the number of mtDNAs had already reset to ~200, indicating that the adjustment occurs within one cell cycle (data added to Figure 3K,L as “mid-L1”).

Reviewer #1 (Recommendations for the authors):– line 177: 'gfp(11)' it took a while for me to realize this is a part of GFP used for BiFC – just because I couldn't tell what '(11)' was referring to (I even thought (11) was referring to reference…). Can you add a bit of explanation so that there will be no confusion? E.g. 'tagged with a fragment of GFP ('GFP(11)') that is used for BiFC assay'.

We clarified the use of split-GFP as a form of BiFC in the text (see pg. 9, lines 207-211). We also amended the text to indicate the GFP subunits contained within each fragment as subscripts, since our prior use of parentheses could be confused with reference citations (Original: GFP(1-10) and GFP(11); New: GFP_1-10_ and GFP_11_).

– the term 'purifying selection' feels a bit too strong for only ~50% reduction in the number and only ~5% reduction in heteroplasmy. It is fine to correlate the current work with purifying selection, but probably better not to call the observation 'purifying selection'.

The use of ‘purifying selection’ to describe the selective reduction of mutant mtDNA is in line with previous publications, including several in *C. elegans* with respect to mild reductions in *uaDf5* mtDNA heteroplasmy (Gitschlag et al., 2020, *eLife* 9: e56686; Ahier et al., 2018, NCB 20: 352-360). Although the consistent reduction in *uaDf5* heteroplasmy we observe in PGCs is modest within one generation, the effect over several generations could be substantial (see Discussion; pg.14 lines 342-345). Unfortunately, this is difficult to test, as *uaDf5* also has a selfish replication advantage (Yang et al., 2022, NCB 24: 181-193), which overcomes *pink-1*-mediated purifying selection in the PGCs and adult germ line since the mutant is inherited at a stable heteroplasmy over many generations (Tsang and Lemire, 2002; Biochem Cell Biol. 80: 645-654). Because of the precedent in the literature, we have opted to keep the term ‘purifying selection’ in the manuscript.

Reviewer #3 (Recommendations for the authors):I have the following recommendations:1. With respect to mtDNA replication in PGCs, this could be addressed experimentally by measuring mtDNA replication using EdU, although this may be technically challenging and not possible in their system. They could address this textually by being clearer throughout the text that they are not measuring replication, and perhaps discussing how, if replication is occurring in PGCs, this would influence their interpretation of their data.

See essential revisions above.

2. In the introduction, consider describing in greater detail worm germline development and cannibalism.

We felt that introducing background on the *C. elegans* PGCs would fit better in the initial section of the Results rather than the Introduction, which focuses on mtDNA and its inheritance. To address Reviewer 3’s point, we expanded on a brief introduction to PGC birth and lobe cannibalism in the first paragraph of the Results (pg. 4, lines 85-90).

3. The Mito-GFP(1-10)PGC + TFAM-GFP(11) looks diffuse and not punctate (see figure S4). Given this, how did the authors count GFP puncta in this strain? Consider including a more detailed explanation of this quantification in the methods.

Mitochondria are highly dynamic, and mtDNAs are in motion within mitochondria. Even in a single frame of laser scanning confocal live imaging there is some slight movement of nucleoids, and later embryos move as well. This limits the spatial resolution of TFAM-GFP live, and could be the cause of diffuse TFAM signal. In order to get a relative measure of TFAM-GFP signal, we defined ‘foci’ as local signal maxima that colocalized with mito-mCherry signal using the ImageJ 3D maxima plugin. We clarified this in the methods (pg. 35, lines 646-652).

4. On lines 296 -297 the authors state "It is possible that this number of mtDNAs is needed for sufficient selection against deleterious mtDNA mutations". Does the fact that selection is not impaired in nop-1 mutants suggest otherwise? If so, consider removing this statement.

Thank you for bringing up this point. Reviewer 3 is correct that the *nop-1* data indicates that cannibalism-based reduction of mtDNAs is not required for selection against *uaDf5* in PGCs. It remains possible, however, that this number is important for the formation of an mtDNA genetic bottleneck. We have amended the text to clarify these points (pg. 14, lines 333-339).